# Do Stochastic, Feel Noiseless: Stable Stochastic Optimization via a Double Momentum Mechanism

**Tehila Dahan**
ECE Department
Technion
Haifa, Israel
t.dahan@campus.technion.ac.il

**Kfir Y. Levy**
ECE Department
Technion
Haifa, Israel
kfirylevy@technion.ac.il

## Abstract

Optimization methods are crucial to the success of machine learning, with Stochastic Gradient Descent (SGD) serving as a foundational algorithm for training models. However, SGD is often sensitive to the choice of the learning rate, which necessitates extensive hyperparameter tuning. In this work, we introduce a new variant of SGD that brings enhanced stability in two key aspects. **First**, our method allows the use of the same fixed learning rate to attain optimal convergence rates **regardless of the noise magnitude**, eliminating the need to adjust learning rates between noiseless and noisy settings. **Second**, our approach achieves these optimal rates over a **wide range of learning rates**, significantly reducing sensitivity compared to standard SGD, which requires precise learning rate selection. Our key innovation is a novel gradient estimator based on a double-momentum mechanism that combines two recent momentum-based techniques. Utilizing this estimator, we design both standard and accelerated algorithms that are robust to the choice of learning rate. Specifically, our methods attain optimal convergence rates in both noiseless and noisy stochastic convex optimization scenarios without the need for learning rate decay or fine-tuning. We also prove that our approach maintains optimal performance across a wide spectrum of learning rates, underscoring its stability and practicality. Empirical studies further validate the robustness and enhanced stability of our approach.

## 1 Introduction

Stochastic Convex Optimization (SCO) is a fundamental framework that captures several classical Machine Learning (ML) problems, such as linear regression, logistic regression, and SVMs (Support Vector Machines), amongst others. In the past two decades, SCO has been extensively explored and highly influenced the field of ML: it popularized the use of Stochastic Gradient Descent (SGD) as the standard workhorse for training ML models; see e.g. Shalev-Shwartz et al. (2007); Welling & Teh (2011); Mairal et al. (2009); Recht et al. (2011); as well as has lead to the design of sophisticated SGD variants that play a central role in training modern large scale models (Duchi et al., 2011; Kingma & Ba, 2015).

One practical difficulty in applying SGD-type methods is the need to tune its learning rate among other hyperparameters, and it is well known that the performance of such algorithms crucially relies on the right choice of the learning rate. Adaptive SGD variants, such as AdaGrad and Adam (Duchi et al., 2011; Kingma & Ba, 2015; Levy et al., 2018; Kavis et al., 2019; Jacobsen & Cutkosky, 2022) have been designed to alleviate this issue by adjusting the learning rate during training. However, despite reducing the need for hyperparameter tuning, adaptive methods can introduce additional complexity and may not always lead to better generalization performance. In many applications, practitioners still prefer to employ standard SGD because it often results in better test error and improved generalization compared to adaptive methods (Wu et al., 2016; Ruder, 2016). Thus, designing SGD variants that are robust to the choice of learning rate, while retaining the simplicity and generalization benefits of standard SGD, can be extremely beneficial in practice.

To address these challenges, we propose a new approach that retains the simplicity and generalization benefits of standard SGD while significantly enhancing its robustness to learning rate selection. Moreover, *since our method focuses on stabilizing the gradient estimation rather than adapting the learning rate, it is orthogonal to adaptive techniques and could potentially be combined with them to yield even better performance.*

**Focusing on the SCO Setting:** In this paper we focus on the prevalent SCO setting where the objective (expected loss) is an Expectation Over Smooth losses (SCO-EOS); this applies e.g. to linear and logistic regression problems (though not to SVMs). In this case, it is well known that SGD requires a careful tuning of the learning rate to obtain the optimal performance. For example, in the noiseless case, SGD (or GD in this case) should employ a learning rate of $\eta^{\text{Offline}} = 1/L$ where $L$ is the smoothness parameter of the objective. Nevertheless, if we apply this $\eta^{\text{Offline}}$ in the noisy setting, the guarantees of SGD become vacuous. To obtain the optimal SGD guarantees, we should roughly decrease the learning rate by a factor of $\sigma\sqrt{T}$ where $T$ is the total number of SGD iterates (and samples), and $\sigma$ is the variance of the noise in the gradient estimates. This illustrates the sensitivity of SGD to the choice of $\eta$, a challenge that also affects stochastic accelerated methods such as those in Lan (2012); Hu et al. (2009); Xiao (2010).

**Contributions.** We introduce a novel gradient estimator for SCO-EOS problems that uses a single sample per-iterate, and shows that its square error, $\|\epsilon_t\|^2$, *shrinks with the number of updates* as $\|\epsilon_t\|^2 \propto 1/t$, where $t$ is the iterate. This, in contrast to the standard SGD estimator where usually $\|\epsilon_t\|^2 = \text{Variance}_t = O(1)$. Our new estimator blends two recent mechanisms that are related to the notion of momentum: Anytime Averaging, which is due to Cutkosky (2019); and a corrected momentum technique (Cutkosky & Orabona, 2019). We therefore denote our estimator by $\mu^2$ which stands for **Momentum²**.

As described below, our new estimator enables to "Do Stochastic (optimization), while feeling Noiseless", i.e. it allows us to use similar machinery as GD employs in the noiseless case. Specifically, **(i)** we can use the exact same fixed learning rate as is used in GD, irrespective of the noise; **(ii)** it enables us to use the norm of the gradient estimates as a stopping criteria, which is a common practice for GD (Beck, 2014). Finally, **(iii)** it enables us to design new SGD variants which are extremely robust to the choice of the learning rate, significantly reducing sensitivity compared to standard SGD.

Concretely, we design an SGD variant called $\mu^2$-SGD, as well as an accelerated version called $\mu^2 - \texttt{ExtraSGD}$, that employs our new estimator and demonstrates their stability with respect to the choice of the learning rates $\eta$. We demonstrate the following,

• **For $\mu^2$-SGD:** Upon using the **exact same learning rate** of $\eta^{\text{Offline}} = 1/8LT$ (where $T$ is the total number of iterates/data-samples), $\mu^2$-SGD enjoys a convergence rate of $O(L/T)$ in the noiseless case, and a rate of $O(L/T + \tilde{\sigma}/\sqrt{T})$ in the noisy case. Moreover, in the noisy case, $\mu^2$-SGD enjoys the same convergence rate as of the optimal SGD $O(L/T + \tilde{\sigma}/\sqrt{T})$, for **a wide range of learning rate choices** i.e. $\eta \in [\eta^{\min}, \eta^{\max}]$, with the ratio $\eta^{\max}/\eta^{\min} \approx (\tilde{\sigma}/L)\sqrt{T}$.

• **For $\mu^2 - \texttt{ExtraSGD}$:** Upon using the **exact same learning rate** of $\eta^{\text{Offline}} = 1/2L$, $\mu^2 - \texttt{ExtraSGD}$ enjoys an optimal convergence rate of $O(L/T^2)$ in the noiseless case, and an optimal rate of $O(L/T^2 + \tilde{\sigma}/\sqrt{T})$ in the noisy case. Moreover, in the noisy case, $\mu^2 - \texttt{ExtraSGD}$ enjoys the same optimal convergence of $O(L/T^2 + \tilde{\sigma}/\sqrt{T})$, **for an *extremely wide* range of learning rate choices** i.e. $\eta \in [\eta^{\min}, \eta^{\max}]$, with the ratio $\eta^{\max}/\eta^{\min} \approx (\tilde{\sigma}/L)T^{3/2}$. The optimal rates mentioned above are also *tight for SCO-EOS problems*, see e.g. Thm. 16.7 in Cutkosky (2022a).

These ratios are substantially larger than the corresponding ratio for standard SGD, where the optimal convergence is achieved only when $\eta^{\max}/\eta^{\min} \approx O(1)$. This establishes the substantial improvement in stability of our approach compared to standard SGD (see Appendix A for a detailed discussion).

We empirically demonstrate the improved stability and performance of our methods over various baselines, confirming both the theoretical and practical advantages of our approach.

On the technical side, it is important to note that individually, each of the momentum techniques that we combine is unable to ensure the stability properties that we are able to ensure for their appropriate combination, i.e. for $\mu^2$-SGD and for $\mu^2 - \texttt{ExtraSGD}$. Moreover, our accelerated version $\mu^2 - \texttt{ExtraSGD}$, requires a careful and delicate blend of several techniques in the right interweaved manner, which leads to a concise yet delicate analysis.

**Related Work:** The Gradient Descent (GD) algorithm and its stochastic counterpart SGD (Robbins & Monro, 1951) are cornerstones of ML and Optimization. Their adoption in various fields has lead to the development of numerous elegant and useful variants (Duchi et al., 2011; Kingma & Ba, 2015; Ge et al., 2015). Curiously, many SGD variants that serve in practical training of non-convex learning models; were originally designed under the framework of SCO.

As we mention in the introduction, the performance of SGD crucially relies on the choice of the learning rate. There is a plethora of work on designing methods that implicitly and optimally adapt the learning rate throughout the learning process (Duchi et al., 2011; Kingma & Ba, 2015; Kavis et al., 2019; Antonakopoulos et al., 2022; Jacobsen & Cutkosky, 2022; Ivgi et al., 2023; Defazio & Mishchenko, 2023); and such methods are widely adopted among practitioners. Nevertheless, in several practical scenarios, standard (non-adaptive) SGD has proven to yields better generalization compared to adaptive variants, albeit still necessitating to find an appropriate learning rate (see e.g. Giladi et al. (2019)).

Momentum (Polyak, 1964) is another widely used practical technique (Sutskever et al., 2013), and it is interestingly related to the accelerated method of Nesterov (Nesterov, 1983) – a seminal approach that enables to obtain faster convergence rates compared to GD for smooth and convex objectives. While Nesterov's accelerated method is fragile to noise, Lan (2012); Hu et al. (2009); Xiao (2010) have designed stochastic accelerated variants that enable to obtain a convergence rate that interpolates between the fast rate in the noiseless case and between the standard SGD rate in the noisy case (depending on the noise magnitude).

Our work builds on two recent mechanisms related to the notion of momentum: **(i)** An Anytime averaging mechanism Cutkosky (2019) which relies on averaging the query points of the gradient oracle. And **(ii)** a corrected momentum technique Cutkosky & Orabona (2019) which relies on averaging the gradients themselves throughout the learning process (while introducing correction). It is interesting to note that the Anytime mechanism has proven to be extremely useful in designing adaptive and accelerated methods (Cutkosky, 2019; Kavis et al., 2019; Antonakopoulos et al., 2022). The corrected momentum mechanism has mainly found use in designing optimal and adaptive algorithms for stochastic non-convex problems (Cutkosky & Orabona, 2019; Levy et al., 2021).

## 2 SETTING

Consider stochastic optimization problems with a convex objective $f : \mathcal{K} \mapsto \mathbb{R}$ that satisfies,

$$f(x) := \mathbf{E}_{z \sim \mathcal{D}} f(x; z) , \tag{1}$$

where $\mathcal{K} \subseteq \mathbb{R}^d$ is a compact convex set, and $\mathcal{D}$ is an unknown distribution from which we may draw i.i.d. samples $\{z_t \sim \mathcal{D}\}_t$. We consider first order optimization methods that iteratively employ such samples in order to generate a sequence of query points and eventually output a solution $x_{\text{output}} \in \mathcal{K}$. Our goal is to approximately minimize $f(\cdot)$, so our performance measure is the expected excess loss,

$$\text{ExcessLoss} := \mathbf{E}[f(x_{\text{output}})] - \min_{x \in \mathcal{K}} f(x) ,$$

where the expectation is w.r.t. the randomization of the samples.

More concretely, at every iteration $t$ such methods maintain a query point $x_t \in \mathbb{R}^d$ which is computed based on the past query points and past samples $\{z_1, \ldots, z_{t-1}\}$. Then, the next query point $x_{t+1}$ is computed based on $x_t$ and on a gradient estimate $g_t$ that is derived by drawing a fresh sample $z_t \sim \mathcal{D}$ independently of past samples, and computing, $g_t := \nabla f(x_t; z_t)$ . Note that this derivative is w.r.t. $x$. The independence between samples implies that $g_t$ is an unbiased estimate of $\nabla f(x_t)$ in the following sense, $\mathbf{E}[g_t | x_t] = \nabla f(x_t)$ . It is often comfortable to think of the computation of $g_t = \nabla f(x_t; z_t)$ as a *(noisy) Gradient Oracle* that upon receiving a query point $x_t \in \mathcal{K}$ outputs a vector $g_t \in \mathbb{R}^d$, which is an unbiased estimate of $\nabla f(x_t)$.

**Assumptions.** We will make the following assumptions,
**Bounded Diameter:** There exists $D > 0$ such: $\max_{x,y \in \mathcal{K}} \|x - y\| \leq D$.
**Bounded variance:** There exists $\sigma > 0$ such,

$$\mathbf{E}\|\nabla f(x; z) - \nabla f(x)\|^2 \leq \sigma^2 , \ \forall x \in \mathcal{K} \tag{2}$$

**Expectation over smooth functions:** There exists $L > 0$ such $\forall x, y \in \mathcal{K}$, $z \in \textbf{Support}\{\mathcal{D}\}$,

$$\|\nabla f(x; z) - \nabla f(y; z)\| \leq L\|x - y\|, \tag{3}$$

This implies that the expected loss $f(\cdot)$ is $L$ smooth.

**Bounded Smoothness Variance.** The above assumption implies that there exists $\sigma_L^2 \in [0, L^2]$ such,

$$\mathbf{E} \left\| (\nabla f(x; z) - \nabla f(x)) - (\nabla f(y; z) - \nabla f(y)) \right\|^2 \leq \sigma_L^2 \|x - y\|^2, \quad \forall x, y \in \mathcal{K}. \tag{4}$$

Clearly, in the deterministic setting where $f(x; z) = f(x)$, $\forall z \in \textbf{Support}\{\mathcal{D}\}$, we have $\sigma_L = 0$. In App. B, we show how Eq. (3) implies Eq. (4).

**Notation:** The notation $\nabla f(x; z)$ relates to gradients with respect to $x$, i.e., $\nabla := \nabla_x$. We use $\|\cdot\|$ to denote the Euclidean norm. Given a sequence $\{y_t\}_t$ we denote $y_{t_1:t_2} := \sum_{\tau=t_1}^{t_2} y_\tau$. For a positive integer $N$ we denote $[N] := \{1, \ldots, N\}$. We also let $\Pi_{\mathcal{K}} : \mathbb{R}^d \mapsto \mathcal{K}$ denote the orthogonal projection onto $\mathcal{K}$, i.e. $\Pi_{\mathcal{K}}(x) = \arg\min_{y \in \mathcal{K}} \|y - x\|^2$, $\forall x \in \mathbb{R}^d$. We shall also denote $\tilde{\sigma}^2 := 32D^2\sigma_L^2 + 2\sigma^2$.

## 3 MOMENTUM MECHANISMS

Here, we provide background regarding two mechanisms that are related to the notion of momentum. Curiously, these approaches are related to averaging of different elements of the learning algorithm. Our approach presented in Sec. 4 builds on a combination of these aforementioned mechanisms.

### 3.1 MECHANISM I: ANYTIME-GD

This first mechanism is related to averaging the **query points** for the noisy gradient oracle. While in standard SGD we query the gradients at the iterates that we compute, in Anytime-SGD (Cutkosky, 2019), we query the gradients at weighted averages of the iterates that we compute.

More concretely, the Anytime-SGD algorithm (Cutkosky, 2019; Kavis et al., 2019) that we describe in Equations (5) and (6), employs a learning rate $\eta > 0$ and a sequence of non-negative weights $\{\alpha_t\}_t$. The algorithm maintains two sequences $\{w_t\}_t, \{x_t\}_t$. At initialization $x_1 = w_1$, and,

$$w_{t+1} = w_t - \eta\alpha_t g_t, \forall t \in [T], \text{ where } g_t = \nabla f(x_t; z_t), \tag{5}$$

where $z_t \sim \mathcal{D}$. Then Anytime-SGD updates,

$$x_{t+1} = \frac{\alpha_{1:t}}{\alpha_{1:t+1}} x_t + \frac{\alpha_{t+1}}{\alpha_{1:t+1}} w_{t+1}. \tag{6}$$

The above implies that the $x_t$'s are weighted averages of the $w_t$'s, i.e. that $x_{t+1} = \frac{1}{\alpha_{1:t+1}} \sum_{\tau=1}^{t+1} \alpha_\tau w_\tau$. Thus, at every iterate, the gradient $g_t$ is queried at $x_t$ which is a weighted average of past iterates, and then $w_{t+1}$ is updated similarly to GD with a weight $\alpha_t$ on the gradient $g_t$.

Curiously, it was shown in Wang et al. (2021) (see Sec. 4.5.1), that a very similar algorithm to Anytime-GD, can be related to the classical Heavy-Ball method (Polyak, 1964), and the latter incorporates momentum in its iterates. Cutkosky (2019) has shown that Anytime-SGD obtains the same convergence rates as SGD for convex loss functions (both smooth and non-smooth). And this technique was found to be extremely useful in designing universal accelerated methods (Cutkosky, 2019; Kavis et al., 2019).

The next theorem is crucial in analyzing Anytime-SGD, and actually applies more broadly,

**Theorem 3.1** (Rephrased from Theorem 1 in Cutkosky (2019)). *Let $f : \mathcal{K} \mapsto \mathbb{R}$ be a convex function with a minimum $w^* \in \arg\min_{w \in \mathcal{K}} f(w)$. Also let $\{\alpha_t \geq 0\}_t$, and $\{w_t \in \mathcal{K}\}_t, \{x_t \in \mathcal{K}\}_t$, such that $\{x_t\}_t$ is an $\{\alpha_t\}_t$ weighted average of $\{w_t\}_t$, i.e. such that $x_1 = w_1$, and for any $t \geq 1$, $x_{t+1} = \frac{1}{\alpha_{1:t+1}} \sum_{\tau=1}^{t+1} \alpha_\tau w_\tau$. Then the following holds for any $t \geq 1$:*
$\alpha_{1:t} (f(x_t) - f(w^*)) \leq \sum_{\tau=1}^{t} \alpha_\tau \nabla f(x_\tau) \cdot (w_\tau - w^*)$.

The above theorem holds for any sequence $\{w_t \in \mathcal{K}\}_t$, and as a private case it holds for the Anytime-GD algorithm. Thus the above Theorem relates the excess loss of a given algorithm that computes the sequences $\{w_t \in \mathcal{K}\}_t, \{x_t \in \mathcal{K}\}_t$ to its weighted regret, $\mathcal{R}_t := \sum_{\tau=1}^{t} \alpha_\tau \nabla f(x_\tau) \cdot (w_\tau - w^*)$.

## 3.2 MECHANISM II: RECURSIVE CORRECTED AVERAGING

This second mechanism is related to averaging the **gradient estimates** that we compute throughout training, which is a common and crucial technique in practical applications. While straightforward averaging might incur bias into the gradient estimates, it was suggested in Cutkosky & Orabona (2019) to add a bias correction mechanism named STORM (STochastic Recursive Momentum). And it was shown that this mechanism leads to a powerful variance reduction effect.

Concretely, STORM maintains an estimate $d_t$ which is a *corrected* weighted average of past stochastic gradients, and then it performs an SGD-style update step,

$$w_{t+1} = w_t - \eta d_t \, . \tag{7}$$

The corrected momentum estimates are updated as follows,

$$d_t = \nabla f(w_t; z_t) + (1 - \beta_t)(d_{t-1} - \nabla f(w_{t-1}, z_t)) \, , \tag{8}$$

for some $\beta_t \in [0, 1]$. The above implies that $\mathbf{E}[d_t] = \mathbf{E}\nabla f(w_t)$. Nevertheless, in general $\mathbf{E}[d_t|w_t] \neq \nabla f(w_t)$, so $d_t$ is conditionally biased (in contrast to standard SGD estimates). Moreover, the choice of the same sample $z_t$ in the two terms of the above expression is crucial for the variance reduction effect.

## 4 OUR APPROACH: DOUBLE MOMENTUM MECHANISM

Our approach is to combine together the two momentum mechanisms that we describe above. Our algorithm is therefore named $\mu^2$-SGD (Momentum$^2$-Stochastic Gradient Descent), and we describe it in Alg. 1. Intuitively, each of these Momentum (averaging) techniques stabilizes the algorithm, and their combination leads to a method that is almost as stable as offline GD. Note that the right combination of these techniques is crucial to obtaining our results, which cannot be achieved by employing only one of these technique without the other.

We first describe algorithm 1, and then present our main result in Theorem 4.1, showing that the error of the gradient estimates of our approach *shrinks* as we progress. Suggesting that we may use the norm of the gradient estimate as a stopping criteria which is a common practice in GD (Beck, 2014). Another benefit is demonstrated in Thm. 4.2,is that $\mu^2$-SGD obtains the same convergence rate as standard SGD, for a very wide range of learning rates (in contrast to SGD).

Next we elaborate on the ingredients of Alg. 1:
**Update rule:** Note that for generality we allow a broad family of update rules in Eq. (10) of Alg. 1. The only constraint on the update rule is that its iterates $\{w_t\}_t$ always belong to $\mathcal{K}$. Later, we will specifically analyze the natural SGD-style update rule,

$$w_{t+1} = \Pi_\mathcal{K}(w_t - \eta \alpha_t d_t) \, . \tag{9}$$

**Momentum Computation:** From Eq. (12) in Alg. 1 we can see that the momentum $d_t$ is updated similarly to the STORM update in Eq. (8), albeit with two main differences: **(i)** first we incorporate importance weights $\{\alpha_t\}_t$ into STORM and recursively update the weighted momentum $\alpha_t d_t$. More importantly **(ii)** we query the noisy gradients at the averages $x_t$'s rather than in the iterates themselves, and the averages (Eq. (11)) are computed in the spirit of Anytime-SGD. These can be seen in the computation of $g_{t+1}$ and $\tilde{g}_t$ which query the gradients at the averages rather than the iterates. Thus, as promised our algorithms combines two different momentum mechanisms.

Next, we present our main result, which shows that Alg. 1 yields estimates with a very small error. The only limitation on the update rule in Eq. (10) is that its iterates $\{w_t\}_t$ always belong to $\mathcal{K}$.

**Theorem 4.1.** *Let $f : \mathcal{K} \mapsto \mathbb{R}$, and assume that $\mathcal{K}$ is convex with diameter $D$, and that the assumption in Equations (2),(3),(4) hold. Then invoking Alg. 1 with $\{\alpha_t = t + 1\}_t$ and $\{\beta_t = 1/\alpha_t\}$, ensures,*

$$\mathbf{E}\|\epsilon_t\|^2 := \mathbf{E}\|d_t - \nabla f(x_t)\|^2 \leq \tilde{\sigma}^2/t \, ,$$

*here $\epsilon_t := d_t - \nabla f(x_t)$, and $\tilde{\sigma}^2 := 32D^2\sigma_L^2 + 2\sigma^2$. ⋆ In App. D.1 we provide high-prob. bounds.*

So according to the above theorem, the overall error of $d_t$ compared to the exact gradient $\nabla f(x_t)$ *shrinks* as we progress. Conversely, in standard SGD (as well as in Anytime-SGD) the expected square error is *fixed*, namely $\mathbf{E}\|g_t - \nabla f(w_t)\|^2 \leq O(\sigma^2)$.

Based Thm. 4.1, we may analyze Alg. 1 with the specific SGD-type update rule presented in Eq. (9).

---

**Algorithm 1** $\mu^2$-SGD

---

Input: #Iterations $T$, initialization $x_1$, $\eta > 0$, weights $\{\alpha_t\}_t$, Corrected Momentum weights $\{\beta_t\}_t$
**Initialize:** set $w_1 = x_1$, draw $z_1 \sim \mathcal{D}$ and set $d_1 = \nabla f(x_1, z_1)$
**for** $t = 1, \ldots, T$ **do**
    **Iterate Update:**
$$\text{Use an update rule to compute } w_{t+1} \in \mathcal{K} \tag{10}$$

   **Query Update** (Anytime-SGD style)**:**
$$x_{t+1} = \frac{\alpha_{1:t}}{\alpha_{1:t+1}} x_t + \frac{\alpha_{t+1}}{\alpha_{1:t+1}} w_{t+1} \ , \tag{11}$$

   **Update Corrected Momentum** (STORM style)**:**
   draw $z_{t+1} \sim \mathcal{D}$, compute $g_{t+1} := \nabla f(x_{t+1}; z_{t+1})$, and $\tilde{g}_t := \nabla f(x_t; z_{t+1})$ and update,
$$d_{t+1} = g_{t+1} + (1 - \beta_{t+1})(d_t - \tilde{g}_t) \tag{12}$$

   **end for**
   **output:** $x_T$

---

**Theorem 4.2** ($\mu^2$-SGD Guarantees). *Let $f : \mathbb{R}^d \mapsto \mathbb{R}$ be a convex function, and assume that $w^* \in \arg\min_{w \in \mathcal{K}} f(w)$ is also its global minimum in $\mathbb{R}^d$. Also, let us make the same assumptions as in Thm. 4.1. Then invoking Alg. 1 with $\{\alpha_t = t + 1\}_t$ and $\{\beta_t = 1/\alpha_t\}_t$, and using the SGD-type update rule (9) with a learning rate $\eta \leq 1/8LT$ inside Eq. (10) of Alg. 1 guarantees,*

$$\mathbf{E}(f(x_T) - f(w^*)) = \mathbf{E}\Delta_T \leq O\left(\frac{D^2}{\eta T^2} + 2\eta\tilde{\sigma}^2 + \frac{4D\tilde{\sigma}}{\sqrt{T}}\right) \ ,$$

*where $\Delta_t := f(x_t) - f(w^*)$, and $\tilde{\sigma}^2 := 32D^2\sigma_L^2 + 2\sigma^2$.*

**Stability of $\mu^2$-SGD.** The above lemma shows that $\mu^2$-SGD obtains the optimal SGD convergence rates for both offline (noiseless) and noisy case with the same choice of fixed learning rate $\eta^{\text{Offline}} = \frac{1}{8LT}$, which does not depend on the noise $\tilde{\sigma}$. This in contrast to SGD, which require either reducing the offline learning rate by a factor of $\sigma\sqrt{T}$; or using sophisticated adaptive learning rates (Duchi et al., 2011; Levy et al., 2018).

Moreover, letting $\eta^{\text{Noisy}} = 1/(8LT + \tilde{\sigma}T^{3/2}/D)$, than it can be seen that in the noisy case our approach enables to employ learning rates in an extremely wide range of $[\eta^{\text{Noisy}}, \eta^{\text{Offline}}]$; and still obtain the same optimal SGD convergence rate. Indeed note that, $\eta^{\text{Offline}}/\eta^{\text{Noisy}} \approx (\tilde{\sigma}/L)T^{1/2}$.

**Comment:** Note that in Theorem 4.2 we assume that the global minimum of $f(\cdot)$ belongs to $\mathcal{K}$. In the next section we present $\mu^2 - \texttt{ExtraSGD}$ — an accelerated version of $\mu^2$-SGD, that does not require this assumption, and enables to obtain accelerated convergence rates as well as better stability.

## 4.1 EXTENSIONS

**Uniformly Small Error.** We have shown that upon using a single sample per-iteration, our approach enables to yields gradient estimates $d_t$'s with a shrinking square error of $O(1/t)$. Nevertheless, if we like to incorporate early stopping, it is desirable to have a uniformly small error of $O(1/T)$ across all iterates. In the appendix we show that this is possible, and only comes at the price of a logarithmic increase in the overall sample complexity. The idea is to incorporate a decaying batch-size into our approach: at round $t$ we suggest to use a batch-size $b_t \propto T/t$. Thus, along $T$ rounds we use a total of $\sum_{t=1}^{T} b_t = O(T \log T)$ samples. This modification allows a uniformly small square error of $O(1/T)$ across all iterates.

**Accurate Estimates of General Operators.** We have shown that our approach enables to yield $O(1/t)$ estimates for gradient estimates of the query point $\{x_t\}_{t \in [T]}$. This can be similarly generalized to yielding $O(1/t)$ estimates for other operators. For example, if we like to estimate the Hessian $\nabla^2 f(x_t)$, and we assume Lipschitz continuous Hessians and bounded variance analogously to Eq. (2),(3),(4), then we can maintain good Hessian estimates in the spirit of Eq. (12)

**Unweighted Variant of $\mu^2$-SGD.** It is natural to ask whether we can employ the more standard

uniform weights $\{\alpha_t = 1\}_{t \in [T]}$, within $\mu^2$-SGD. Going along very similar lines to our proof for Thm. 4.2, we show that this indeed can be done while using $\eta \propto 1/L \log T$, and yields similar bounds, albeit suffering logarithmic factors in $T$.

## 4.2 NECESSITY OF BOTH MECHANISMS

Here we discuss the importance of combining both Anytime and STORM mechanisms to obtaining the shrinking error property (see Thm. 4.1) of $\mu^2$-SGD, which is a key to our other results. Concretely, the Anytime mechanism (without STORM ) appearing in Equations (5) (6), is employing gradient estimates of the form $g_t := \nabla f(x_t; z_t)$, and it is therefore immediate that error of these gradient estimates $\|\epsilon_t^{\text{Anytime}}\|^2 := \|g_t - \nabla f(x_t)\|^2$ is $O(1)$, which is similar to standard SGD. Conversely, the STORM mechanisms (without Anytime) appearing in Equations (7) (8) maintains estimates $d_t$. The STORM update rule yields a variance reduction, which depends on the distance between consecutive query points i.e. $\|w_t - w_{t-1}\|^2$, which will in turn depend on the learning rate (as in the original STORM paper). This couples between the variance reduction mechanism and the learning rate, and therefore fails to achieve robustness to the learning rate (for example: using a fixed learning rate within the standard STORM approach will fail to converge). Thus, the combination of these techniques is crucial towards obtaining the shrinking error substantiated in Thm. 4.1, which is independent of the learning rate, and therefore allows to obtain stability as we substantiate in Theorems 4.2 and 5.2.

## 4.3 PROOF SKETCH OF THM. 4.1

*Proof.* First note that the $x_t$'s always belong to $\mathcal{K}$, since they are weighted averages of $\{w_t \in \mathcal{K}\}_t$'s. Our first step is to bound the difference between consecutive queries. The definition of $x_t$ implies: $\alpha_{1:t-1}(x_t - x_{t-1}) = \alpha_t(w_t - x_t)$ , yielding,

$$\|x_t - x_{t-1}\|^2 = (\alpha_t/\alpha_{1:t-1})^2 \|w_t - x_t\|^2 \leq (16/\alpha_{t-1}^2)D^2 . \tag{13}$$

where we used $\alpha_t = t+1$ implying $\alpha_t/\alpha_{1:t-1} \leq 4/\alpha_{t-1}$ for any $t \geq 2$, we also used $\|w_t - x_t\| \leq D$. **Notation:** Prior to going on with the proof we shall require some notation. We will denote $\bar{g}_t := \nabla f(x_t)$, and recall the following notation from Alg. 1: $g_t := \nabla f(x_t, z_t)$ ; $\tilde{g}_{t-1} := \nabla f(x_{t-1}, z_t)$. We will also denote, $\epsilon_t := d_t - \bar{g}_t$ .

Recalling Eq. (12), and combining it with the above definition of $\epsilon_t$ enables to derive the following,

$$\alpha_t \epsilon_t = \beta_t \alpha_t(g_t - \bar{g}_t) + (1 - \beta_t)(\alpha_t Z_t + \frac{\alpha_t}{\alpha_{t-1}} \alpha_{t-1} \epsilon_{t-1}) ,$$

where we denote $Z_t := (g_t - \bar{g}_t) - (\tilde{g}_{t-1} - \bar{g}_{t-1})$. Now, using $\alpha_t = t+1$, and $\beta_t = 1/(t+1)$ then it can be shown that $\alpha_t \beta_t = 1$, and $\alpha_t(1 - \beta_t) = \alpha_t - 1 := \alpha_{t-1}$. Moreover, $(1 - \beta_t)\frac{\alpha_t}{\alpha_{t-1}} = \frac{\alpha_t - 1}{\alpha_{t-1}} = 1$. Plugging these above yields,

$$\alpha_t \epsilon_t = \alpha_{t-1} Z_t + \alpha_{t-1} \epsilon_{t-1} + (g_t - \bar{g}_t) = M_t + \alpha_{t-1} \epsilon_{t-1} . \tag{14}$$

where for any $t > 1$ we denote $M_t := \alpha_{t-1} Z_t + (g_t - \bar{g}_t)$, as well as $M_1 = g_1 - \bar{g}_1$. Unrolling the above equation yields an explicit expression for any $t \in [T]$: $\alpha_t \epsilon_t = \sum_{\tau=1}^{t} M_\tau$ .

Notice that the sequence $\{M_t\}_t$ is martingale difference sequence with respect to the natural filtration $\{\mathcal{F}_t\}_t$ induced by the history of the samples up to time $t$; which implies,

$$\mathbf{E}\|\alpha_t \epsilon_t\|^2 = \left\|\sum_{\tau=1}^{t} M_\tau\right\|^2 = \sum_{\tau=1}^{t} \mathbf{E}\|M_\tau\|^2 \leq 2\sum_{\tau=1}^{t} \alpha_{t-1}^2 \mathbf{E}\|Z_t\|^2 + 2\sum_{\tau=1}^{t} \mathbf{E}\|g_t - \bar{g}_t\|^2 . \tag{15}$$

Using Eq. (4) together with Eq. (13) allows to bound, $\mathbf{E}\|Z_t\|^2 = \mathbf{E}\|(g_t - \bar{g}_t) - (\tilde{g}_{t-1} - \bar{g}_{t-1})\|^2 \leq \sigma_L^2 \|x_t - x_{t-1}\|^2 \leq 16\sigma_L^2 D^2/\alpha_{t-1}^2$ , and we may also bound $\mathbf{E}\|g_t - \bar{g}_t\|^2 \leq \sigma^2$. Plugging it above back into Eq. (15) and summing establishes the theorem. $\square$

## 5 ACCELERATED VERSION: $\mu^2 - \text{EXTRASGD}$

Here we present an accelerated version that makes use of a double momentum mechanism as we do for $\mu^2$-SGD. Our approach relies on an algorithmic template named ExtraGradient (Korpelevich,

1976; Nemirovski, 2004; Juditsky et al., 2011). The latter technique has already been combined with the Anytime-SGD mechanism in Cutkosky (2019); Kavis et al. (2019), showing that it leads to acceleration. Here, we further blend an appropriate STORM Mechanism, leading to a new method that we name $\mu^2 - \texttt{ExtraSGD}$. Our main result is presented in Thm. 5.2.

On the technical side, our accelerated version requires a careful and delicate blend of several techniques in the right interweaved manner, which leads to a concise yet delicate analysis.

**Optimistic OGD, Extragradient and UnixGrad:** The extragradient technique is related to an algorithmic template named Optimistic Online GD (Optimistic OGD) (Rakhlin & Sridharan, 2013). In this algorithm we receive a sequence of (possibly arbitrary) loss vectors $\{d_t \in \mathbb{R}^d\}_{t \in [T]}$ in an online manner. And our goal is to compute a sequence of iterates (or decision points) $\{w_t \in \mathcal{K}\}_t$, where $\mathcal{K}$ is given convex set. Note that we may pick $w_t$ only based on past information $\{d_1, \ldots, d_{t-1}\}$. And our goal is to ensure a low weighted regret for any $w \in \mathcal{K}$, where the latter is defined as,

$$\mathcal{R}_T(w) := \sum_{t=1}^{T} \alpha_t d_t \cdot (w_t - w^*) ,$$

and $\{\alpha_t > 0\}$ is a sequence of predefined weights. In the optimistic setting we assume that we may access a sequence of "hint vectors" $\{\hat{d}_t \in \mathbb{R}^d\}_t$ and that prior to picking $w_t$ we may also access $\{\hat{d}_1, \ldots, \hat{d}_t\}$. Rakhlin & Sridharan (2013) have shown that if the hints are useful, in the sense that $\hat{d}_t \approx d_t$, then one can reduce the regret by properly incorporating the hints. Thus, in **Optimistic-OGD** we maintain two sequences: a decision point sequence $\{w_t\}_t$ and an auxiliary sequence $\{y_t\}$, updated as follows,

**Optimistic OGD:**
$$w_t = \arg\min_{w \in \mathcal{K}} \alpha_t \hat{d}_t \cdot w + \frac{1}{2\eta}\|w - y_{t-1}\|^2 \quad \& \quad y_t = \arg\min_{y \in \mathcal{K}} \alpha_t d_t \cdot y + \frac{1}{2\eta}\|y - y_{t-1}\|^2 \quad (16)$$

It was shown in Rakhlin & Sridharan (2013); Kavis et al. (2019) that the above algorithm enjoys the following regret bound for any $w \in \mathcal{K}$,

**Theorem 5.1** (See e.g. the proof of Thm. 1 in Kavis et al. (2019)). *Let $\eta > 0$, $\{\alpha_t \geq 0\}_{t \in [T]}$ and $\mathcal{K}$ be a convex set with bounded diameter $D$. Then Optimistic-OGD ensures the following for any $w \in \mathcal{K}$,*
$$\sum_{t=1}^{T} \alpha_t d_t \cdot (w_t - w) \leq \frac{4D^2}{\eta} + \frac{\eta}{2}\sum_{t=1}^{T}\alpha_t^2\|d_t - \hat{d}_t\|^2 - \frac{1}{2\eta}\sum_{t=1}^{T}\|w_t - y_{t-1}\|^2 .$$

The **Extragradient algorithm** (Nemirovski, 2004) aims to minimize a convex function $f : \mathcal{K} \mapsto \mathbb{R}$. To do so, it applies the Optimistic-OGD template with the following choices of loss and hint vectors: $\hat{d}_t = \nabla f(y_{t-1})$, and $d_t := \nabla f(w_t)$.
The **UnixGrad algorithm** (Kavis et al., 2019), can be seen as an Anytime version of Extragradient, where again we aim to minimize a convex function $f : \mathcal{K} \mapsto \mathbb{R}$. In the spirit of Anytime-GD, UnixGrad maintains two sequences of *weighted averages* $\{x_t, \hat{x}_t\}_t$ based on $\{w_t, y_t\}_t$,

$$\hat{x}_t = \frac{\alpha_{1:t-1}}{\alpha_{1:t}}x_{t-1} + \frac{\alpha_t}{\alpha_{1:t}}y_{t-1} , \ x_t = \frac{\alpha_{1:t-1}}{\alpha_{1:t}}x_{t-1} + \frac{\alpha_t}{\alpha_{1:t}}w_t \quad (17)$$

Then, based on the above averages UnixGrad sets the loss and hint vectors as follows: $\hat{d}_t = \nabla f(\hat{x}_t)$, and $d_t := \nabla f(x_t)$. Note that the above averaging rule implies that the $x_t$'s are weighted averages of the $w_t$'s, i.e. $x_t = \frac{1}{\alpha_{1:t}}\sum_{\tau=1}^{t}\alpha_\tau w_\tau$. The latter enables to utilize the Anytime guarantees of Thm. 3.1.

There also exist stochastic versions of the above approaches where we may only query noisy gradients.

**Our Approach.** We suggest to employ the Optimistic-OGD template together with a STORM mechanism on top of the Anytime mechanism employed by UnixGrad. Specifically, we maintain the same weighted averages as in Eq. (17), and define momentum estimates as follows: At round $t$ draw a fresh sample $z_t \sim \mathcal{D}$, and compute

$$\tilde{g}_{t-1} = \nabla f(x_{t-1}; z_t) \ , \ \hat{g}_t = \nabla f(\hat{x}_t; z_t) \ , \ g_t = \nabla f(x_t; z_t) . \quad (18)$$

Based on the above compute the (corrected momentum) loss and hint vectors as follows,

$$\alpha_t \hat{d}_t = \alpha_t \hat{g}_t + (1 - \beta_t)\alpha_t(d_{t-1} - \tilde{g}_{t-1}) \ \& \ \alpha_t d_t = \alpha_t g_t + (1 - \beta_t)\alpha_t(d_{t-1} - \tilde{g}_{t-1}) \quad (19)$$

---

**Algorithm 2** $\mu^2 - \texttt{ExtraSGD}$

---

Input: #Iterations $T$, initialization $y_0$, $\eta > 0$, weights $\{\alpha_t\}_t$, Corrected Momentum weights $\{\beta_t\}_t$

**Initialize:** set $x_0 = 0$, and $\hat{x}_1 = y_0$, draw $z_1 \sim \mathcal{D}$ and set $d_0 = \tilde{g}_0 = \hat{d}_1 = \nabla f(\hat{x}_1, z_1)$

**for** $t = 1, \ldots, T$ **do**

    **Compute:** $w_t = \arg\min_{w \in \mathcal{K}} \alpha_t \hat{d}_t \cdot w + \frac{1}{2\eta} \|w - y_{t-1}\|^2$ , **& Update:** $x_t = \frac{\alpha_{1:t-1}}{\alpha_{1:t}} x_{t-1} + \frac{\alpha_t}{\alpha_{1:t}} w_t$

    **Compute:** $g_t = \nabla f(x_t; z_t)$ **& Update:** $d_t = g_t + (1 - \beta_t)(d_{t-1} - \tilde{g}_{t-1})$

    **Compute:** $y_t = \arg\min_{y \in \mathcal{K}} \alpha_t d_t \cdot y + \frac{1}{2\eta} \|y - y_{t-1}\|^2$ **& Update:** $\hat{x}_{t+1} = \frac{\alpha_{1:t}}{\alpha_{1:t+1}} x_t + \frac{\alpha_{t+1}}{\alpha_{1:t+1}} y_t$

    **Draw** a fresh sample $z_{t+1} \sim \mathcal{D}$ and compute, $\tilde{g}_t = \nabla f(x_t; z_{t+1})$ , $\hat{g}_{t+1} = \nabla f(\hat{x}_{t+1}, z_{t+1})$

    **Update:** $\hat{d}_{t+1} = \hat{g}_{t+1} + (1 - \beta_{t+1})(d_t - \tilde{g}_t)$

**end for**
**output:** $x_T$

---

And then update according to Optimistic OGD in Eq. (16). Notice that the update rule for $\alpha_t d_t$ is the exact same update that we use in Alg. 1; additionally as we have already commented, the $x_t$ sequence in Eq. (17) is an $\{\alpha_t\}_t$ weighted average of the $\{w_t\}_t$ sequence. Therefore, if we pick $\alpha_t = t + 1$, and $\beta_t = 1/\alpha_t$, then we can invoke Thm. 4.1 implying that,

$$\mathbf{E}\|\epsilon_t\|^2 := \mathbf{E}\|d_t - \nabla f(x_t)\|^2 \leq \tilde{\sigma}^2/t , \quad \text{where} \quad \tilde{\sigma}^2 := 32D^2\sigma_L^2 + 2\sigma^2 .$$

The pseudo-code in Alg. 2 depicts our $\mu^2 - \texttt{ExtraSGD}$ algorithm with the appropriate computational order. It can be seen that it combines Optimistic-OGD updates (Eq. (16)), together with appropriate Anytime averaging (Eq. (17)), and together with STORM updates for $d_t, \hat{d}_t$ (Eq. (19)).

We are now ready to state the guarantees of $\mu^2 - \texttt{ExtraSGD}$ ,

**Theorem 5.2** ($\mu^2 - \texttt{ExtraSGD}$ ). *Let $f : \mathcal{K} \mapsto \mathbb{R}$ be a convex function and $\mathcal{K}$ a convex set with diameter $D$, and denote $w^* \in \arg\min_{w \in \mathcal{K}} f(w)$. Then under the assumption in Equations (2),(3),(4), invoking Alg. 2 with $\{\alpha_t = t + 1\}_t$ and $\{\beta_t = 1/\alpha_t\}_t$, and $\eta \leq 1/2L$ guarantees,*

$$\mathbf{E}(f(x_T) - f(w^*)) := \mathbf{E}\Delta_T \leq O\left(\frac{D^2}{\eta T^2} + \frac{\tilde{\sigma}D}{\sqrt{T}}\right) .$$

As can be seen in Alg. 2, the appropriate $\mu^2$ accelerated version requires a careful and delicate blend of the aforementioned techniques in the right interweaved manner.

**Stability of $\mu^2 - $ ExtraSGD .** The above lemma shows that $\mu^2 - \texttt{ExtraSGD}$ obtains the optimal rates for both offline (noiseless) and noisy cases with the same choice of fixed learning rate $\eta^{\text{Offline}} = 1/2L$. This contrasts existing accelerated methods, which require either to reduce the offline learning rate by a factor of $\sigma\sqrt{T}$ (Xiao, 2010); or to employ sophisticated adaptive learning rates (Cutkosky, 2019; Kavis et al., 2019). Moreover, letting $\eta^{\text{Noisy}} := 1/(2L + \tilde{\sigma}T^{3/2}/D)$, then it can be seen that in the noisy case, our approach enables to employ learning rates in **an extremely wide range** of $[\eta^{\text{Noisy}}, \eta^{\text{Offline}}]$; and still obtain the same optimal convergence rate of $O\left(LD^2/T^2 + \tilde{\sigma}D/\sqrt{T}\right)$. Indeed note that, the ratio $\eta^{\text{Offline}}/\eta^{\text{Noisy}} \approx (\tilde{\sigma}/L)T^{3/2}$. **Moreover**, conversely to Thm. 4.2, which requires $w^* \in \arg\min_{w \in \mathcal{K}} f(w)$ to be also the *global minimum* of $f(\cdot)$; Thm. 5.2 does not require this assumption.

## 6 EXPERIMENTS

We begin by evaluating our proposed $\mu^2$-SGD algorithm in a *convex* setting, where model weights were projected onto a unit ball after each gradient update. The evaluation is conducted on the MNIST dataset (LeCun et al., 2010), using a logistic regression model. We compare our approach with the parameters suggested by our theoretical framework ($\alpha_t = t$, $\beta_t = \frac{1}{t}$) against several baseline optimizers. This includes each individual component of the $\mu^2$-SGD algorithm—STORM and AnyTime-SGD—all tested with the same parameter settings. As illustrated in Figure 1 the $\mu^2$-SGD algorithm consistently demonstrates superior stability across a wide range of learning rates. Notably,

while the AnyTime-SGD algorithm maintains strong stability over a broad spectrum of learning rates, STORM encounters difficulties at higher rates. By combining both methods, $\mu^2$-SGD achieves greater stability and performance, outperforming each approach individually. Additionally, when considering a more typical range of learning rates for this setup (see Figure 1b), $\mu^2$-SGD may not always achieve the absolute best result compared the other algorithms. Nevertheless, it consistently achieves high performance and robustness across a broader range of learning rates, significantly reducing the need for an extensive search to find a high-performing learning rate. On top of that, by leveraging the momentum parameters grounded in our theoretical framework, $\mu^2$-SGD further eliminates the need for hyperparameter tuning—a process that can be highly computationally expensive.

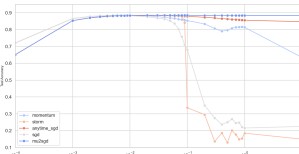 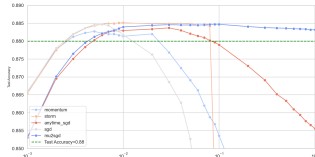

a Over a Wide Range of Leaning Rates.    b Over a Typical Range of Learning Rates.

Figure 1: MNIST: Test Accuracy Over Different Learning Rates in a Convex Setting (↑ is better).

**Deep Learning Variant.** We demonstrate the effectiveness of our approach in *non-convex* settings using a 2-layer convolutional network on the MNIST dataset and ResNet-18 on the CIFAR-10 dataset (Krizhevsky et al., 2014). First, we reformulate the AnyTime update, originally defined as $x_t := \frac{\alpha_t w_t + \alpha_{1:t-1} x_{t-1}}{\alpha_{1:t}}$ into a mathematically equivalent momentum-based approach:

$$x_t = \gamma_t w_t + (1 - \gamma_t) x_{t-1}$$

where $\gamma_t := \frac{\alpha_t}{\alpha_{1:t}}$. For non-convex models, decaying momentum parameters in the iteration number can be overly aggressive; thus, we propose a heuristic approach using fixed momentum parameters to improve adaptability. Note that, by setting $\alpha_t = C\alpha_{1:t-1}$, where $C > 0$ is a constant, we derive $\gamma_t = \frac{C}{C+1}$, making it fixed for all time steps $t \geq 1$.

We show that using fixed momentum parameters ($\gamma_t = 0.1$, $\beta_t = 0.9$) in the non-convex setting ensures *high stability* and *strong performance* **(i)** across a wide range of learning rates and **(ii)** over random seeds, as shown in Figure 2 and in App. H.3. Consistent results were observed on both MNIST and CIFAR-10, as detailed in App. H.3. These findings highlight the robustness and adaptability of our method, making it a reliable choice for optimizing non-convex models.

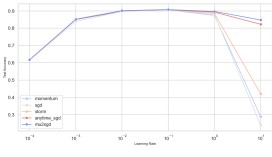 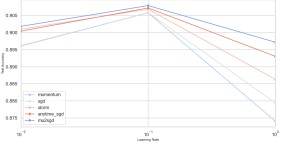 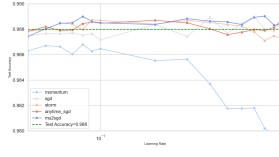

a CIFAR-10: Over a Wide Range.    b CIFAR-10: Over a Typical Search Range.    c MNIST: Over a Typical Search Range.

Figure 2: Test Accuracy Over a Range of Learning Rates in Non-Convex Setups (↑ is better).

All experiments were conducted using the PyTorch framework. The convex experiments were run on an Apple M2 chip, while the non-convex experiments were executed on an NVIDIA A30 GPU. The results were averaged over three different random seeds. For additional details, refer to Appendix H and our GitHub repository.[1]

# 7    CONCLUSION

By carefully blending two recent momentum techniques, we designed a new shrinking-error gradient estimate for the SCO-EOS setting. Based on it, we presented two algorithms that rely on SGD and Extragradient templates and showed their significant stability w.r.t. the choice of the learning rate, thus enabling a much more robust training. In the future, it will be interesting to further explore the applicability of our non-convex heuristic for huge-scale models, which require much more computational resources. Moreover, it will be interesting to understand whether we can design an algorithm for non-convex problems, with similar theoretical properties to our approach.

---

[1] https://github.com/dahan198/mu2sgd

## ACKNOWLEDGEMENT

This research was partially supported by Israel PBC-VATAT, by the Technion Artificial Intelligent Hub (Tech.AI) and by the Israel Science Foundation (grant No. 3109/24).

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

# A   STABILITY W.R.T. CHOICE OF LEARNING RATE

Here we formally explain what we mean when we relate to the stability of an algorithm $\mathcal{A}$ w.r.t. choice of the learning rate $\eta$.

Given a SCO algorithm $\mathcal{A}$, we can usually present its generalization bounds as follows:
For a given choice of $0 \leq \eta \leq \bar{\eta}$, then the algorithm $\mathcal{A}$ ensures,

$$\textbf{Excess-Loss} \leq \mathcal{R}_{\mathcal{A}}(\eta) \ ,$$

where $\mathcal{R} : \mathbb{R}_+ \mapsto \mathbb{R}_+$ is the convergence rate as a function of the learning rate $\eta$. And this description applies to all the methods that we mention in our paper.

In this case we can define the optimal learning rate as follows,

$$\eta^* := \min_{\eta \in [0,\bar{\eta}]} \mathcal{R}_{\mathcal{A}}(\eta) \ .$$

And we define the **range of order optimal learning rates of** $\mathcal{A}$ to be:

$$\mathrm{Range}_{\mathcal{A}} := \{\eta \in [0,\bar{\eta}] : \mathrm{R}_{\mathcal{A}}(\eta) \leq 2\mathrm{R}_{\mathcal{A}}(\eta^*)\} \ .$$

In words, this set is comprised of all learning rates that achieve the optimal convergence rate up to a multiplicative factor of $2$ [2].

Usually, the set $\mathrm{Range}_{\mathcal{A}}$ is a line segment in $\mathbb{R}_+$, and we can therefore write $\mathrm{Range}_{\mathcal{A}} = [\eta^{\min}, \eta^{\max}]$. And we can further denote,

$$\textbf{ratio}_{\mathcal{A}} = \eta^{\max}/\eta^{\min} \ .$$

Thus, higher ratios imply improved stability of $\mathcal{A}$ w.r.t. choice of the learning rate $\eta$. Next, we compare the stability for the methods that we mention in our paper, for the SCO-EOS setting that we describe in Sec. 2. And substantiate the improved stability of $\mu^2$-SGD and of $\mu^2 - \texttt{Extra}$SGD over standard SGD and accelerated stochastic SGD.

**Stability of Standard SGD:**   For standard SGD it is well known that for the choice $\eta \in (0, \frac{1}{2L}]$ it enjoys a convergence rate of,

$$\mathcal{R}_{\mathrm{SGD}}(\eta) := \frac{D^2}{\eta T} + \eta \sigma^2$$

Thus, in the typical case where $\frac{D}{\sigma \sqrt{T}} \leq \frac{1}{2L}$ (i.e. when the noise is not negligible) we have $\eta^* = \frac{D}{\sigma \sqrt{T}}$, and $\mathcal{R}_{\mathrm{SGD}}(\eta^*) = \frac{2D\sigma}{\sqrt{T}}$. And it can therefore be validated that,

$$\textbf{ratio}_{\mathrm{SGD}} \leq 15 \ .$$

Conversely, **(ii)** in the non typical case where $\frac{D}{\sigma \sqrt{T}} > \frac{1}{2L}$ we have $\eta^* = \frac{1}{2L}$. In this case we can validate again that,

$$\textbf{ratio}_{\mathrm{SGD}} \leq 15 \ .$$

This substantiates that for standard SGD we have stability ratio of $\textbf{ratio}_{\mathrm{SGD}} \approx O(1)$.

**Stability of Accelerated Stochastic SGD:**   There are several variants of accelerated stochastic SGD. For such algorithms, and for a learning rate choice of $\eta \in (0, \frac{1}{2L}]$, such methods enjoy a convergence rate of (See e.g. Theorem 2 in Lan (2012)),

$$\mathcal{R}_{\mathrm{Accel-SGD}}(\eta) := \frac{D^2}{\eta T^2} + \eta \sigma^2 T$$

Thus, similarly to our analysis of standard SGD, it can be validated that for such methods we have, $\textbf{ratio}_{\mathrm{Accel-SGD}} \approx O(1)$.

---

[2]The choice of 2 is rather arbitrary, and we can similarly choose any factor sufficiently greater than 1.

**Stability of $\mu^2$-SGD:** As we establish in Theorem 4.2 our $\mu^2$-SGD approach ensures that for the choice of $\eta \in (0, \frac{1}{8LT}]$, it enjoys a convergence rate of,

$$\mathcal{R}_{\mu^2-\text{SGD}}(\eta) := \frac{D^2}{\eta T^2} + 2\eta\tilde{\sigma}^2 + \frac{4D\tilde{\sigma}}{\sqrt{T}} \ .$$

Thus, in this case we have $\eta^* = \frac{D}{\sqrt{2}\tilde{\sigma}T}$, and $\mathcal{R}_{\mu^2-\text{SGD}}(\eta^*) = \frac{2\sqrt{2}D\tilde{\sigma}}{T} + \frac{4D\tilde{\sigma}}{\sqrt{T}}$. And it can therefore be validated that in this case we have $\eta^{\min} \approx \frac{D}{\tilde{\sigma}T^{3/2}}$ and $\eta^{\max} = \frac{1}{8LT}$, and therefore,

$$\textbf{ratio}_{\mu^2-\text{SGD}} \approx \frac{\tilde{\sigma}}{LD}\sqrt{T} \ .$$

**Stability of $\mu^2 - \texttt{ExtraSGD}$:** As we establish in Theorem 5.2 our $\mu^2 - \texttt{ExtraSGD}$ approach ensures that for the choice of $\eta \in (0, \frac{1}{2L}]$, it enjoys a convergence rate of,

$$\mathcal{R}_{\mu^2 - \texttt{ExtraSGD}}(\eta) := \frac{D^2}{\eta T^2} + \frac{\tilde{\sigma}D}{\sqrt{T}} \ .$$

Thus, in this case we have $\eta^* = \frac{1}{2L}$, and $\mathcal{R}_{\mu^2 - \texttt{ExtraSGD}}(\eta^*) = \frac{2LD^2}{T^2} + \frac{\tilde{\sigma}D}{\sqrt{T}}$. And it can therefore be validated that in this case we have $\eta^{\min} \approx \frac{D}{\tilde{\sigma}T^{3/2}}$ and $\eta^{\max} = \frac{1}{2L}$, and therefore,

$$\textbf{ratio}_{\mu^2 - \texttt{ExtraSGD}} \approx \frac{\tilde{\sigma}}{LD}T^{3/2} \ .$$

## B EXPLAINING THE BOUNDED SMOOTHNESS VARIANCE ASSUMPTION

Here we show that Eq. (3) implies that Eq. (4) holds for some $\sigma_L^2 \in [0, L]$.

Fixing $x, y \in \mathcal{K}$, then Eq. (3) implies that for any $z \in \textbf{Support}\{\mathcal{D}\}$ there exists $L_{x,y;z} \in [0, L]$ such that,

$$\|\nabla f(x; z) - \nabla f(y; z)\|^2 = L_{x,y;z}^2 \|x - y\|^2 \ .$$

Similarly there exists $L_{x,y} \in [0, L]$ such that,

$$\|\nabla f(x) - \nabla f(y)\|^2 = L_{x,y}^2 \|x - y\|^2 \ .$$

And clearly in the deterministic case we have $L_{x,y;z} = L_{x,y} \ , \forall z \in \textbf{Support}\{\mathcal{D}\}$. Therefore,

$$\mathbf{E}\|(\nabla f(x; z) - \nabla f(x)) - (\nabla f(y; z) - \nabla f(y))\|^2 = \mathbf{E}\|\nabla f(x; z) - \nabla f(y; z)\|^2 - \|\nabla f(x) - \nabla f(y)\|^2$$
$$= \mathbf{E}(L_{x,y;z}^2 - L_{x,y}^2) \cdot \|x - y\|^2 = \sigma_L^2\{x, y\} \cdot \|x - y\|^2 \ ,$$

where we have used $\mathbf{E}(\nabla f(x; z) - \nabla f(y; z)) = (\nabla f(x) - \nabla f(y))$, and we denote $\sigma_L^2\{x, y\} := \mathbf{E}(L_{x,y;z}^2 - L_{x,y}^2)$. This notation implies that $\sigma_L^2\{x, y\} \in [0, L^2]$, and clearly $\sigma_L^2\{x, y\} = 0$ in the deterministic case for all $x, y \in \mathcal{K}$. Thus, if we denote,

$$\sigma_L^2 := \sup_{x,y \in \mathcal{K}} \sigma_L^2\{x, y\} \ ,$$

Then $\sigma_L^2 \in [0, L^2]$ satisfies Eq. (4) and is equal to 0 in the deterministic (noiseless) case.

## C PROOF OF THM. 4.1

*Proof of Thm. 4.1.* First note that the $x_t$'s always belong to $\mathcal{K}$, since they are weighted averages of the $\{w_t \in \mathcal{K}\}_t$'s. Next we bound the difference between consecutive queries. By definition,

$$\alpha_{1:t-1}(x_t - x_{t-1}) = \alpha_t(w_t - x_t) \ ,$$

Implying,

$$\|x_t - x_{t-1}\|^2 = (\alpha_t/\alpha_{1:t-1})^2 \|w_t - x_t\|^2 \leq (16/t^2)D^2 = (16/\alpha_{t-1}^2)D^2 \ . \tag{20}$$

where we have used $\alpha_t = t+1$ implying $\alpha_t/\alpha_{1:t-1} \le 4/t$ for any $t \ge 2$, we also used $\|w_t - x_t\| \le D$ which holds since $w_t, x_t \in \mathcal{K}$, finally we use $\alpha_{t-1} = t$.

**Notation:** Prior to going on with the proof we shall require some notation. We will denote $\bar{g}_t := \nabla f(x_t)$, and recall the following notation form Alg. 1: $g_t := \nabla f(x_t, z_t)$ ; $\tilde{g}_{t-1} := \nabla f(x_{t-1}, z_t)$, and we will also denote, $\bar{g}_t := \nabla f(x_t)$ ,and

$$\epsilon_t := d_t - \bar{g}_t .$$

Now, recalling Eq. (12),

$$\alpha_t d_t = \alpha_t g_t + (1 - \beta_t)\alpha_t(d_{t-1} - \tilde{g}_{t-1}) .$$

Combining the above with the definition of $\epsilon_t$ yields the following recursive relation,

$$
\begin{aligned}
\alpha_t \epsilon_t &:= \alpha_t d_t - \alpha_t \bar{g}_t \\
&= \alpha_t(g_t - \bar{g}_t) + (1 - \beta_t)\alpha_t(d_{t-1} - \tilde{g}_{t-1}) \\
&= \beta_t \alpha_t(g_t - \bar{g}_t) + (1 - \beta_t)\alpha_t(d_{t-1} - \tilde{g}_{t-1} + g_t - \bar{g}_t) \\
&= \beta_t \alpha_t(g_t - \bar{g}_t) + (1 - \beta_t)\alpha_t(\bar{g}_{t-1} - \tilde{g}_{t-1} + g_t - \bar{g}_t) + (1 - \beta_t)\alpha_t(d_{t-1} - \bar{g}_{t-1}) \\
&= \beta_t \alpha_t(g_t - \bar{g}_t) + (1 - \beta_t)\alpha_t((g_t - \bar{g}_t) - (\tilde{g}_{t-1} - \bar{g}_{t-1})) + (1 - \beta_t)\alpha_t(d_{t-1} - \bar{g}_{t-1}) \\
&= \beta_t \alpha_t(g_t - \bar{g}_t) + (1 - \beta_t)\alpha_t Z_t + (1 - \beta_t)\frac{\alpha_t}{\alpha_{t-1}}\alpha_{t-1}\epsilon_{t-1}
\end{aligned}
$$

where we denoted $Z_t := (g_t - \bar{g}_t) - (\tilde{g}_{t-1} - \bar{g}_{t-1})$. Now, using $\alpha_t = t+1$, and $\beta_t = 1/(t+1)$ then it can be shown that $\alpha_t \beta_t = 1$,and $\alpha_t(1 - \beta_t) = \alpha_t - 1 := \alpha_{t-1}$. Moreover, $(1 - \beta_t)\frac{\alpha_t}{\alpha_{t-1}} = \frac{\alpha_t - 1}{\alpha_{t-1}} = 1$. Using these relations in the equation above gives,

$$\alpha_t \epsilon_t = \alpha_{t-1}Z_t + \alpha_{t-1}\epsilon_{t-1} + (g_t - \bar{g}_t) = M_t + \alpha_{t-1}\epsilon_{t-1} . \tag{21}$$

where for any $t > 1$ we denote $M_t := \alpha_{t-1}Z_t + (g_t - \bar{g}_t)$, as well as $M_1 = g_1 - \bar{g}_1$. Unrolling the above equation yields an explicit expression for any $t \in [T]$,

$$\alpha_t \epsilon_t = \sum_{\tau=1}^{t} M_\tau . \tag{22}$$

Now, notice that the sequence $\{M_t\}_t$ is is martingale difference sequence with respect to the natural filtration $\{\mathcal{F}_t\}_t$ induced by the history of the samples up to time $t$. Indeed,

$$\mathbf{E}[M_t|\mathcal{F}_{t-1}] = \mathbf{E}[(g_t - \bar{g}_t)|\mathcal{F}_{t-1}] + \alpha_{t-1}\mathbf{E}[Z_t|\mathcal{F}_{t-1}] = \mathbf{E}[(g_t - \bar{g}_t)|x_t] + \alpha_{t-1}\mathbf{E}[Z_t|x_{t-1}, x_t] = 0 .$$

Thus, using Lemma C.1 below gives,

$$
\begin{aligned}
\mathbf{E}\|\alpha_t \epsilon_t\|^2 &= \left\|\sum_{\tau=1}^{t} M_\tau\right\|^2 = \sum_{\tau=1}^{t} \mathbf{E}\|M_\tau\|^2 = \sum_{\tau=1}^{t} \mathbf{E}\|\alpha_{t-1}Z_t + (g_t - \bar{g}_t)\|^2 \\
&\le 2\sum_{\tau=1}^{t} \alpha_{t-1}^2 \mathbf{E}\|Z_t\|^2 + 2\sum_{\tau=1}^{t} \mathbf{E}\|g_t - \bar{g}_t\|^2 \\
&\le 2\sum_{\tau=1}^{t} \alpha_{t-1}^2 \mathbf{E}\|(g_t - \bar{g}_t) - (\tilde{g}_{t-1} - \bar{g}_{t-1})\|^2 + 2\sum_{\tau=1}^{t} \sigma^2 \\
&= 2\sum_{\tau=1}^{t} \alpha_{t-1}^2 \mathbf{E}\|(\nabla f(x_t; z_t) - \nabla f(x_t)) - (\nabla f(x_{t-1}; z_t) - \nabla f(x_{t-1}))\|^2 + 2t\sigma^2 \\
&\le 2\sum_{\tau=1}^{t} \alpha_{t-1}^2 \sigma_L^2 \|x_t - x_{t-1}\|^2 + 2t\sigma^2 \\
&\le 32D^2 \sigma_L^2 \sum_{\tau=1}^{t} (\alpha_{t-1}^2/\alpha_{t-1}^2) + 2t\sigma^2 \\
&= (32D^2 \sigma_L^2 + 2\sigma^2) \cdot t \\
&= \tilde{\sigma}^2 \cdot t .
\end{aligned}
\tag{23}
$$

here the first inequality uses $\|a + b\|^2 \leq 2\|a\|^2 + 2\|b\|^2$ which holds for any $a, b \in \mathbb{R}^d$; the second inequality uses the bounded variance assumption; the third inequality uses Eq. (4), and the last inequality uses Eq. (20).

Dividing the above inequality by $\alpha_t^2 = (t+1)^2$ the lemma follows,

$$\mathbf{E}\|d_t - \nabla f(x_t)\|^2 = \mathbf{E}\|\epsilon_t\|^2 = \mathbf{E}\|\alpha_t \epsilon_t\|^2 / \alpha_t^2 \leq \tilde{\sigma}^2 t/(t+1^2) \leq \tilde{\sigma}^2/(t+1) .$$

**Lemma C.1.** *Let $\{M_t\}_t$ be a martingale difference sequence with respect to a filtration $\{\mathcal{F}_t\}_t$, then the following holds for any $t$,*

$$\mathbf{E}\left\|\sum_{\tau=1}^{t} M_\tau\right\|^2 = \sum_{\tau=1}^{t} \mathbf{E}\left\|M_\tau\right\|^2 .$$

$\square$

## C.1 Proof of Lemma C.1

*Proof of Lemma C.1.* We shall prove the lemma by induction over $t$. The base case where $t = 1$ clearly holds.

Now for induction step let us assume that the equality holds for $t \geq 1$ and lets prove it holds for $t + 1$. Indeed,

$$\begin{aligned}
\mathbf{E}\left\|\sum_{\tau=1}^{t+1} M_\tau\right\|^2 &= \mathbf{E}\left\|M_{t+1} + \sum_{\tau=1}^{t} M_\tau\right\|^2 \\
&= \mathbf{E}\left\|\sum_{\tau=1}^{t} M_\tau\right\|^2 + \mathbf{E}\|M_{t+1}\|^2 + 2\mathbf{E}\left(\sum_{\tau=1}^{t} M_\tau\right) \cdot M_{t+1} \\
&= \sum_{\tau=1}^{t+1} \mathbf{E}\left\|M_\tau\right\|^2 + 2\mathbf{E}\left[\mathbf{E}\left[\left(\sum_{\tau=1}^{t} M_\tau\right) \cdot M_{t+1}|\mathcal{F}_t\right]\right] \\
&= \sum_{\tau=1}^{t+1} \mathbf{E}\left\|M_\tau\right\|^2 + 2\mathbf{E}\left[\left(\sum_{\tau=1}^{t} M_\tau\right) \cdot \mathbf{E}\left[M_{t+1}|\mathcal{F}_t\right]\right] \\
&= \sum_{\tau=1}^{t+1} \mathbf{E}\left\|M_\tau\right\|^2 + 0 \\
&= \sum_{\tau=1}^{t+1} \mathbf{E}\left\|M_\tau\right\|^2 ,
\end{aligned}$$

where the third line follows from the induction hypothesis, as well as from the law of total expectations; the fourth lines follows since $\{M_\tau\}_{\tau=1}^{t}$ are measurable w.r.t $\mathcal{F}_t$, and the fifth line follows since $\mathbf{E}[M_{t+1}|\mathcal{F}_t] = 0$. Thus, we have established the induction step and therefore the lemma holds. $\square$

## D Extensions

Here we provide several extensions and additions to Theorem 4.1.

- In Sec. D.1 we provide high-probability bounds for $\|\epsilon_t\|^2$.
- In Sec. D.2 we show how to obtain $\mathbf{E}\|\epsilon_t\|^2 \leq O(1/T)$ at the price of additional $O(\log T)$ factor in the total sample complexity (i.e. we show that this requires a total of $O(T \log T)$ samples rather than $O(T)$ samples).
- Our extension to a more standard variant of SGD, which employs uniform weights and a learning rate of $\eta \propto 1/L \log T$ appears in Sec. G

### D.1 High Probability Bounds

To obtain high probability bounds we shall require another assumption, that the stochastic gradients in $\mathcal{K}$ are bounded, i.e. that the exist $G > 0$ such $\|\nabla f(x,z)\| \leq G$, $\forall x \in \mathcal{K}, z \in \textbf{Support}\{\mathcal{D}\}$. We are now ready to show that w.p. $\geq 1 - \delta$ then for all $t \in [T]$ we have,

$$\|\epsilon_t\|^2 \leq O\left(\frac{\tilde{\sigma}^2}{t} \cdot \log(T/\delta) + \frac{U_{\max}^2}{t^2} \cdot \log(T/\delta)\right) ,$$

where we denote $U_{\max} := 8LD + 2G$.

Recall that in the proof of Theorem 4.1 we show the following in Eq. (22),

$$\alpha_t \epsilon_t = \sum_{\tau=1}^{t} M_\tau := M_{1:t} . \tag{24}$$

where $M_t := \alpha_{t-1} Z_t + (g_t - \bar{g}_t)$, as well as $M_1 = g_1 - \bar{g}_1$, where we denoted $Z_t := (g_t - \bar{g}_t) - (\tilde{g}_{t-1} - \bar{g}_{t-1})$. Thus $M_t$ is a martingale difference sequence w.r.t. the natural filtration induced by the upcoming samples. And $M_t$ is also bounded w.p. 1 since,

$$\begin{aligned}
\|M_t\| &= \|\alpha_{t-1} Z_t + (g_t - \bar{g}_t)\| \\
&\leq \|\alpha_{t-1} Z_t\| + \|(g_t - \bar{g}_t)\| \\
&\leq \|\alpha_{t-1}(g_t - \tilde{g}_{t-1})\| + \|\alpha_{t-1}(\bar{g}_t - \bar{g}_{t-1})\| + \|(g_t - \bar{g}_t)\| \\
&\leq L\alpha_{t-1}\|x_t - x_{t-1}\| + L\alpha_{t-1}\|x_t - x_{t-1}\| + 2G \\
&\leq 2L\alpha_{t-1} \cdot 4D/\alpha_{t-1} + 2G \\
&= 8LD + 2G \\
&:= U_{\max} .
\end{aligned}$$

where we have used the smoothness of the $\nabla f(\cdot, z)$ as well as Eq. (20). We also denote $U_{\max} := 8LD + 2G$.

Finally, similarly to the proof of Theorem 4.1 we can show that,

$$\mathbf{E}_{t-1}\|M_t\|^2 = \tilde{\sigma}^2$$

where $\mathbf{E}_{t-1}$ denotes conditional expectation conditioned over history of the samples (randomizations) up until and including round $t - 1$.

Now, since the $\{M_t\}_t$ is a martingale sequence w.r.t. the natural filtration induced by optimization process, and since it is bounded, with bounded conditional second moments, then we can use Cor. 4.1 in Minsker (2017) [3] to show that w.p. $\geq 1 - \delta$ the for all $t \in [T]$ we have,

$$\|M_{1:t}\|^2 \leq O\left(\tilde{\sigma}^2 t \cdot \log(T/\delta) + U_{\max}^2 \log(T/\delta)\right)$$

where the $T$ inside the logarithm comes from using the union bound.

Thus, based on the above and on Eq. (24), we immediately conclude that, w.p. $\geq 1 - \delta$ then for all $t \in [T]$ we have,

$$\|\epsilon_t\|^2 = \frac{1}{\alpha_t^2}\|M_{1:t}\|^2 \leq O\left(\frac{\tilde{\sigma}^2}{t} \cdot \log(T/\delta) + \frac{U_{\max}^2}{t^2} \cdot \log(T/\delta)\right)$$

where we used $\alpha_t = t$. This concludes the proof.

### D.2 Uniformly Small Error Bounds

Here we show that upon increasing the sample complexity by a factor of $\log T$, enables to obtain a uniformly small bound of $\mathbf{E}\|\epsilon_t\|^2 \leq \tilde{\sigma}^2/T$, $\forall t \in [T]$.

To do do, we will employ a batch-size of size $b_t = \lceil T/t \rceil$ at round $t$.

---

[3]Actaully Cor. 4.1 in Minsker (2017) is a Corollary of Thm. 3.1 therein, which applies to a sum of independent matrices. Nevertheless, we can obtain a corollary for the Martingale difference case for vectors from Thm. 3.2 in Minsker (2017) which applies to this martingale difference case.

**Total # Samples.** The total number of samples that we use along all rounds is therefore $\sum_{t=1}^{T} b_t = O(T \log T)$. Thus the sample complexity only increases by $\log T$ factor.

**Error Analysis.** Upon using a batch-size $b_t$ the variance of our estimator in round $t$ decreases by a factor of $b_t$. Thus, along the exact same lines as in Eq. (23) we can show the following.

$$\mathbf{E}\|\alpha_t \epsilon_t\|^2 \leq \sum_{\tau=1}^{t} \frac{\tilde{\sigma}^2}{b_\tau} \leq \frac{\tilde{\sigma}^2}{T} \sum_{\tau=1}^{t} \tau \leq \tilde{\sigma}^2 \frac{t^2}{T} \, .$$

Recalling $\alpha_t = t$ and dividing by $\alpha_t^2$ yields,

$$\mathbf{E}\|\epsilon_t\|^2 \leq \frac{\tilde{\sigma}^2}{T} \, ,$$

which establishes the uniformly small error.

# E  PROOF OF THM. 4.2

*Proof of Thm. 4.2.* The proof is a direct combination of Thm. 4.1 together with the standard regret bound of OGD (Online Gradient Descent), which in turn enables to utilize the Anytime guarantees of Thm. 3.1.

**Part 1: Regret Bound.** Standard regret analysis of the update rule in Eq. (9) implies the following for any $t$ (see e.g. (Hazan et al., 2016), as well as Theorem 15.1 in (Cutkosky, 2022b)),

$$\sum_{\tau=1}^{t} \alpha_\tau d_\tau \cdot (w_\tau - w^*) \leq \frac{D^2}{2\eta} + \frac{\eta}{2} \sum_{\tau=1}^{t} \alpha_\tau^2 \|d_\tau\|^2 \, . \tag{25}$$

**Part 2: Anytime Guarantees.** Since the $x_t$'s are weighted averages of the $w_t$'s we may invoke Thm. 3.1, which ensures for any $t \in [T]$,

$$\alpha_{1:t} \Delta_t = \alpha_{1:t}(f(x_t) - f(w^*)) \leq \sum_{\tau=1}^{t} \alpha_\tau \nabla f(x_\tau) \cdot (w_\tau - w^*) \, ,$$

where we denote $\Delta_t := f(x_t) - f(w^*)$.

**Part 3: Combining Guarantees.** Combining the above Anytime guarantees together with the bound in Eq. (25) yields,

$$\begin{aligned}
\alpha_{1:t} \Delta_t &\leq \sum_{\tau=1}^{t} \alpha_\tau \nabla f(x_\tau) \cdot (w_\tau - w^*) \\
&= \sum_{\tau=1}^{t} \alpha_\tau d_\tau \cdot (w_\tau - w^*) + \sum_{\tau=1}^{t} \alpha_\tau (\nabla f(x_\tau) - d_\tau) \cdot (w_\tau - w^*) \\
&= \frac{D^2}{2\eta} + \frac{\eta}{2} \sum_{\tau=1}^{t} \alpha_\tau^2 \|d_\tau\|^2 - \sum_{\tau=1}^{t} \alpha_\tau \epsilon_\tau \cdot (w_\tau - w^*) \\
&\leq \frac{D^2}{2\eta} + \frac{\eta}{2} \sum_{\tau=1}^{t} \alpha_\tau^2 \|\nabla f(x_\tau) + \epsilon_\tau\|^2 + \sum_{\tau=1}^{t} \|\alpha_\tau \epsilon_\tau\| \cdot \|w_\tau - w^*\| \\
&\leq \frac{D^2}{2\eta} + \eta \sum_{\tau=1}^{t} \alpha_\tau^2 \|\nabla f(x_\tau)\|^2 + \eta \sum_{\tau=1}^{t} \alpha_\tau^2 \|\epsilon_\tau\|^2 + D \sum_{\tau=1}^{t} \|\alpha_\tau \epsilon_\tau\| \\
&\leq \frac{D^2}{2\eta} + 2\eta L \sum_{\tau=1}^{t} \alpha_\tau^2 \Delta_\tau + \eta \sum_{\tau=1}^{t} \alpha_\tau^2 \|\epsilon_\tau\|^2 + D \sum_{\tau=1}^{t} \|\alpha_\tau \epsilon_\tau\| \\
&\leq \frac{D^2}{2\eta} + 4\eta L \sum_{\tau=1}^{t} \alpha_{1:\tau} \Delta_\tau + \eta \sum_{\tau=1}^{t} \alpha_\tau^2 \|\epsilon_\tau\|^2 + D \sum_{\tau=1}^{t} \|\alpha_\tau \epsilon_\tau\| \, ,
\end{aligned} \tag{26}$$

where the first inequality follows from Cauchy-Schwartz; the second inequality holds since $\|w_t - w^*\| \leq D$, as well as from using $\|a + b\|^2 \leq 2\|a\|^2 + 2\|b\|^2$ which holds for any $a, b \in \mathbb{R}^d$, the third inequality follows by the self bounding property for smooth functions (see Lemma E.1 below) implying that $\|\nabla f(x_\tau)\|^2 \leq 2L(f(x_\tau) - f(w^*)) := 2L\Delta_\tau$; and the fourth inequality follows due to $\alpha_\tau^2 \leq 2\alpha_{1:\tau}$ which holds since $\alpha_\tau = \tau + 1$.

**Lemma E.1.** *(See e.g. (Levy et al., 2018; Cutkosky, 2019)) Let $F : \mathbb{R}^d \mapsto \mathbb{R}$ be an L-smooth function with a global minimum $x^*$, then for any $x \in \mathbb{R}^d$ we have,*

$$\|\nabla F(x)\|^2 \leq 2L(F(x) - F(w^*)) .$$

Next, we will take expectation over Eq. (26), yielding,

$$
\begin{aligned}
\alpha_{1:t}\mathbf{E}\Delta_t &\leq \frac{D^2}{2\eta} + 4\eta L\sum_{\tau=1}^{t}\alpha_{1:\tau}\mathbf{E}\Delta_\tau + \eta\sum_{\tau=1}^{t}\alpha_\tau^2\mathbf{E}\|\epsilon_\tau\|^2 + D\sum_{\tau=1}^{t}\mathbf{E}\|\alpha_\tau\epsilon_\tau\| \\
&\leq \frac{D^2}{2\eta} + 4\eta L\sum_{\tau=1}^{t}\alpha_{1:\tau}\mathbf{E}\Delta_\tau + \eta\sum_{\tau=1}^{t}\alpha_\tau^2\mathbf{E}\|\epsilon_\tau\|^2 + D\sum_{\tau=1}^{t}\sqrt{\alpha_\tau^2\mathbf{E}\|\epsilon_\tau\|^2} \\
&\leq \frac{D^2}{2\eta} + 4\eta L\sum_{\tau=1}^{t}\alpha_{1:\tau}\mathbf{E}\Delta_\tau + \eta\sum_{\tau=1}^{t}\alpha_\tau^2\cdot\tilde{\sigma}^2/\alpha_\tau + D\sum_{\tau=1}^{t}\sqrt{\alpha_\tau^2\cdot\tilde{\sigma}^2/\alpha_\tau} \\
&\leq \frac{D^2}{2\eta} + 4\eta L\sum_{\tau=1}^{t}\alpha_{1:\tau}\mathbf{E}\Delta_\tau + \eta\tilde{\sigma}^2\sum_{\tau=1}^{t}\alpha_\tau + D\tilde{\sigma}\sum_{\tau=1}^{t}\sqrt{\alpha_\tau} \\
&\leq \frac{D^2}{2\eta} + 4\eta L\sum_{\tau=1}^{T}\alpha_{1:\tau}\mathbf{E}\Delta_\tau + \eta\tilde{\sigma}^2\sum_{\tau=1}^{T}\alpha_\tau + D\tilde{\sigma}\sum_{\tau=1}^{T}\sqrt{\alpha_\tau} \\
&\leq \frac{D^2}{2\eta} + 4\eta L\sum_{\tau=1}^{T}\alpha_{1:\tau}\mathbf{E}\Delta_\tau + \eta\tilde{\sigma}^2\alpha_{1:T} + 2D\tilde{\sigma}T^{3/2} \\
&\leq \frac{D^2}{2\eta} + \frac{1}{2T}\sum_{\tau=1}^{T}\alpha_{1:\tau}\mathbf{E}\Delta_\tau + \eta\tilde{\sigma}^2\alpha_{1:T} + 2D\tilde{\sigma}T^{3/2} ,
\end{aligned}
\tag{27}
$$

where the second lines is due to Jensen's inequality implying that $\mathbf{E}X \leq \sqrt{\mathbf{E}X^2}$ for any random variable $X$; the third line follows from $\mathbf{E}\|\epsilon_t\|^2 \leq \tilde{\sigma}^2/\alpha_t$ which holds by Thm. 4.1; the fifth line holds since $t \leq T$; the sixth line follows since $\sum_{t=1}^{T}\sqrt{\alpha_t} \leq 2T^{3/2}$, and the last line follows since we pick $\eta \leq 1/8LT$.

To obtain the final bound we will apply the lemma below to Eq. (27),

**Lemma E.2.** *Let $\{A_t\}_{t\in[T]}$ be a sequence of non-negative elements and $\mathcal{B} \in \mathbb{R}$, and assume that for any $t \leq T$,*

$$A_t \leq \mathcal{B} + \frac{1}{2T}\sum_{t=1}^{T}A_t ,$$

*Then the following bound holds,*

$$A_T \leq 2\mathcal{B} .$$

Taking $A_t \leftarrow \alpha_{1:t}\mathbf{E}\Delta_t$ and $\mathcal{B} \leftarrow \frac{D^2}{2\eta} + \eta\tilde{\sigma}^2\alpha_{1:T} + 2D\tilde{\sigma}T^{3/2}$ provides the following explicit bound,

$$\alpha_{1:T}\mathbf{E}\Delta_T \leq \frac{D^2}{\eta} + 2\eta\tilde{\sigma}^2\alpha_{1:T} + 4D\tilde{\sigma}T^{3/2}$$

Dividing by $\alpha_{1:T}$ and recalling $\alpha_{1:T} = \Theta(T^2)$ gives,

$$\mathbf{E}(f(x_T) - f(w^*)) = \mathbf{E}\Delta_T \leq O\left(\frac{D^2}{\eta T^2} + 2\eta\tilde{\sigma}^2 + \frac{4D\tilde{\sigma}}{\sqrt{T}}\right) ,$$

which concludes the proof. $\qquad\square$

### E.1 PROOF OF LEMMA E.2

*Proof of Lemma E.2.* Summing the inequality $A_t \leq \mathcal{B} + \frac{1}{2T}\sum_{t=1}^{T} A_t$ over $t$ gives,

$$A_{1:T} \leq T\mathcal{B} + T\frac{1}{2T}A_{1:T} = T\mathcal{B} + \frac{1}{2}A_{1:T} \,,$$

Re-ordering we obtain,

$$A_{1:T} \leq 2T\mathcal{B} \,.$$

Plugging this back to the original inequality and taking $t = T$ gives,

$$A_T \leq \mathcal{B} + \frac{1}{2T}A_{1:T} \leq 2\mathcal{B} \,.$$

which concludes the proof. $\qquad\square$

## F PROOF OF THM. 5.2

*Proof of Thm. 5.2.* The proof decomposes according to the techniques that $\mu^2 - \texttt{ExtraSGD}$ employs.

**Part I: Anytime Guarantees.** Since the $x_t$'s are $\{\alpha_t\}_t$ weighted averages of the $\{w_t\}_t$'s we can invoke Thm. 3.1 which implies,

$$\alpha_{1:T}(f(x_T) - f(w^*)) \leq \sum_{t=1}^{T} \alpha_t \nabla f(x_t) \cdot (w_t - w^*) = \sum_{t=1}^{T} \alpha_t d_t \cdot (w_t - w^*) - \sum_{t=1}^{T} \alpha_t \epsilon_t \cdot (w_t - w^*) \,.$$
(28)

where we have denote $\epsilon_t := d_t - \nabla f(x_t)$.

**Part II: Optimistic OGD Guarantees.** Since the update rule for $\{w_t, y_t\}_t$ satisfies the Optimistic-OGD template w.r.t the sequences of loss and hint vectors $\{d_t, \hat{d}_t\}_t$ we can apply Lemma 5.1 to bound the weighted regret in Eq. (28) as follows,

$$\alpha_{1:T}(f(x_T) - f(w^*))$$

$$\leq \frac{4D^2}{\eta} + \frac{\eta}{2}\sum_{t=1}^{T} \alpha_t^2 \|d_t - \hat{d}_t\|^2 - \frac{1}{2\eta}\sum_{t=1}^{T} \|w_t - y_{t-1}\|^2 - \sum_{t=1}^{T} \alpha_t \epsilon_t \cdot (w_t - w^*)$$

$$\leq \frac{4D^2}{\eta} + \frac{\eta}{2}\sum_{t=1}^{T} \alpha_t^2 \|g_t - \hat{g}_t\|^2 - \frac{1}{2\eta}\sum_{t=1}^{T} \|w_t - y_{t-1}\|^2 + \sum_{t=1}^{T} \|\alpha_t \epsilon_t\| \cdot \|w_t - w^*\|$$

$$\leq \frac{4D^2}{\eta} + \frac{\eta}{2}\sum_{t=1}^{T} \alpha_t^2 \|\nabla f(x_t; z_t) - \nabla f(\hat{x}_t; z_t)\|^2 - \frac{1}{2\eta}\sum_{t=1}^{T} \|w_t - y_{t-1}\|^2 + D\sum_{t=1}^{T} \|\alpha_t \epsilon_t\|$$

$$\leq \frac{4D^2}{\eta} + \frac{\eta L^2}{2}\sum_{t=1}^{T} \alpha_t^2 \|x_t - \hat{x}_t\|^2 - \frac{1}{2\eta}\sum_{t=1}^{T} \|w_t - y_{t-1}\|^2 + D\sum_{t=1}^{T} \|\alpha_t \epsilon_t\|$$

$$\leq \frac{4D^2}{\eta} + \frac{\eta L^2}{2}\sum_{t=1}^{T} \alpha_t^2 \left(\frac{\alpha_t}{\alpha_{1:t}}\right)^2 \|w_t - y_{t-1}\|^2 - \frac{1}{2\eta}\sum_{t=1}^{T} \|w_t - y_{t-1}\|^2 + D\sum_{t=1}^{T} \|\alpha_t \epsilon_t\|$$

$$\leq \frac{4D^2}{\eta} + \frac{4\eta L^2}{2}\sum_{t=1}^{T} \|w_t - y_{t-1}\|^2 - \frac{1}{2\eta}\sum_{t=1}^{T} \|w_t - y_{t-1}\|^2 + D\sum_{t=1}^{T} \|\alpha_t \epsilon_t\|$$

$$\leq \frac{4D^2}{\eta} + D\sum_{t=1}^{T} \|\alpha_t \epsilon_t\| \,,$$
(29)

where the first line uses Eq. (28) together with Thm. 5.1; the second line uses $d_t - \hat{d}_t = g_t - \hat{g}_t$ which follows by Eq. (19); and the third line follows by the definitions of $g_t, \hat{g}_t$, as well as from $\|w_t - w^*\| \leq D$, which holds since $w_t, w^* \in \mathcal{K}$; the fourth line follows by our assumption in Eq. (3);

the fifth line holds since $x_t - \hat{x}_t = (\alpha_t/\alpha_{1:t})(w_t - y_{t-1})$ which holds due to Eq. (17); the sixth line holds since $\alpha_t^4/(\alpha_{1:t})^2 \leq 4$ , $\forall t \geq 1$; and the last line follows since $2\eta L^2 - 1/(2\eta) \leq 0$ which holds since we assume $\eta \leq 1/2L$.

**Part III: $\mu^2$ Guarantees.** Notice that our definitions for $w_t, x_t, \alpha_t, \beta_t$ and $d_t$ satisfy the exact same conditions of Thm. 4.1, This immediately implies that $\mathbf{E}\|\epsilon_t\|^2 \leq \tilde{\sigma}^2/t$ , $\forall t$. Using this, and taking the expectation of Eq. (29) yields,

$$\alpha_{1:T}\mathbf{E}(f(x_T) - f(w^*)) \leq \frac{4D^2}{\eta} + D\sum_{t=1}^{T}\mathbf{E}\|\alpha_t\epsilon_t\| \leq \frac{4D^2}{\eta} + D\sum_{t=1}^{T}\sqrt{\mathbf{E}\|\alpha_t\epsilon_t\|^2}$$

$$\leq \frac{4D^2}{\eta} + D\tilde{\sigma}\sum_{t=1}^{T}\sqrt{\alpha_t^2/t} \leq \frac{4D^2}{\eta} + 2T^{3/2}D\tilde{\sigma} . \tag{30}$$

where the second inequality uses Jensen's Inequality: $\mathbf{E}X \leq \sqrt{\mathbf{E}X^2}$ which holds for any random variable $X$; the last inequality follows from $\alpha_t^2/t \leq 2t$, implying that $\sum_{t=1}^{T}\sqrt{\alpha_t^2/t} \leq 2T^{3/2}$.

Dividing the above equation by $\alpha_{1:T}$ and recalling that $\alpha_{1:T} = \Theta(T^2)$ concludes the proof. $\quad\square$

# G EXTENSION OF $\mu^2$-SGD TO UNIFORM WEIGHTS AND $\eta \propto 1/L\log T$

Here we show that we can obtain the same guarantees as in Thm. 4.2, when using the following more standard choices of $\alpha_t = 1$, and $\eta \propto 1/L\log T$ inside Alg. 1; albeit suffering $\log T$ factors in the convergence rate.

The next theorem, which is a variant of Thm. 4.1, shows that even upon choosing uniform weights we get $\mathbf{E}\|\epsilon_t\|^2 \leq O(\tilde{\sigma}^2/t)$.

**Theorem G.1.** *Let $f : \mathcal{K} \mapsto \mathbb{R}$, and assume that $\mathcal{K}$ is convex with diameter $D$, and that the assumption in Equations (2),(3),(4) hold. Then invoking Alg. 1 with $\{\alpha_t = 1\}_t$ and $\{\beta_t = 1/t\}$, ensures,*

$$\mathbf{E}\|\epsilon_t\|^2 := \mathbf{E}\|d_t - \nabla f(x_t)\|^2 \leq \tilde{\sigma}^2/t ,$$

*where $\epsilon_t := d_t - \nabla f(x_t)$, and $\tilde{\sigma}^2 := 32D^2\sigma_L^2 + 2\sigma^2$.*

Next we provide a proof sketch. The exact proof follows same lines as the proof of Thm. 4.1.

*Proof Sketch of Thm. G.1.* First note that the $x_t$'s always belong to $\mathcal{K}$, since they are weighted averages of $\{w_t \in \mathcal{K}\}_t$'s. Our first step is to bound the difference between consecutive queries. The definition of $x_t$ implies,

$$\alpha_{1:t-1}(x_t - x_{t-1}) = \alpha_t(w_t - x_t) ,$$

yielding,

$$\|x_t - x_{t-1}\|^2 = (\alpha_t/\alpha_{1:t-1})^2\|w_t - x_t\|^2 \leq \frac{1}{(t-1)^2}D^2 . \tag{31}$$

where we have used $\alpha_t = 1$ and $\alpha_{1:t-1} = t - 1$; we also used $\|w_t - x_t\| \leq D$ which holds since $w_t, x_t \in \mathcal{K}$.

**Notation:** Prior to going on with the proof we shall require some notation. We will denote $\bar{g}_t := \nabla f(x_t)$, and recall the following notation from Alg. 1: $g_t := \nabla f(x_t, z_t)$ ;$\tilde{g}_{t-1} := \nabla f(x_{t-1}, z_t)$. We will also denote, $\epsilon_t := d_t - \bar{g}_t$ .

Now, recalling Eq. (12), using $\alpha_t = 1$, and combining it with the above definition of $\epsilon_t$ enables to derive the following recursive relation,

$$\epsilon_t = \beta_t(g_t - \bar{g}_t) + (1 - \beta_t)Z_t + (1 - \beta_t)\epsilon_{t-1}$$

$$= \frac{1}{t}(g_t - \bar{g}_t) + \frac{t-1}{t}Z_t + \frac{t-1}{t}\epsilon_{t-1} ,$$

where we denote $Z_t := (g_t - \bar{g}_t) - (\tilde{g}_{t-1} - \bar{g}_{t-1})$, and used $\beta_t = 1/t$. Now, multiplying the above equation by $t$ gives,

$$t\epsilon_t = (g_t - \bar{g}_t) + (t-1)Z_t + (t-1)\epsilon_{t-1}$$

$$= M_t + (t-1)\epsilon_{t-1} .$$

where for any $t > 1$ we denote $M_t := (t-1)Z_t + (g_t - \bar{g}_t)$, as well as $M_1 = g_1 - \bar{g}_1$. Unrolling the above equation yields an explicit expression for any $t \in [T]$:

$$t\epsilon_t = \sum_{\tau=1}^{t} M_\tau := M_{1:t} .$$

Noticing that the sequence $\{M_t\}_t$ is is martingale difference sequence with respect to the natural filtration $\{\mathcal{F}_t\}_t$ induced by the history of the samples up to time $t$; enables to bound as follows,

$$\mathbf{E}\|t\epsilon_t\|^2 = \left\|\sum_{\tau=1}^{t} M_\tau\right\|^2 = \sum_{\tau=1}^{t} \mathbf{E}\|M_\tau\|^2 = \sum_{\tau=1}^{t} \mathbf{E}\|(t-1)Z_t + (g_t - \bar{g}_t)\|^2$$

$$\leq 2\sum_{\tau=1}^{t}(t-1)^2 \mathbf{E}\|Z_t\|^2 + 2\sum_{\tau=1}^{t} \mathbf{E}\|g_t - \bar{g}_t\|^2$$

$$\leq 2\sum_{\tau=1}^{t}(t-1)^2 \mathbf{E}\|(g_t - \bar{g}_t) - (\tilde{g}_{t-1} - \bar{g}_{t-1})\|^2 + 2\sum_{\tau=1}^{t} \sigma^2 . \tag{32}$$

Now, using Eq. (4) together with Eq. (31) allows to bound,

$$\mathbf{E}\|(g_t - \bar{g}_t) - (\tilde{g}_{t-1} - \bar{g}_{t-1})\|^2 \leq \sigma_L^2 \|x_t - x_{t-1}\|^2 \leq \sigma_L^2 D^2/(t-1)^2 .$$

Plugging the above back into Eq. (32) and summing establishes the theorem. $\square$

$\mu^2$**-SGD with** $\eta \propto 1/L \log T$   Based on the above theorem we are now ready to state the guarantees A version of $\mu^2$-SGD that employs standard choices of $\alpha_t = 1$ and $\eta \propto 1/L \log T$.

**Theorem G.2** ($\mu^2$-SGD Guarantees). *Let* $f : \mathbb{R}^d \mapsto \mathbb{R}$ *be a convex function, and assume that* $w^* \in \arg\min_{w \in \mathcal{K}} f(w)$ *is also its global minimum in* $\mathbb{R}^d$. *Also, let us make the same assumptions as in Thm. 4.1. Then invoking Alg. 1 with* $\{\alpha_t = 1\}_t$ *and* $\{\beta_t = 1/t\}_t$, *and using the SGD-type update rule* (9) *with a learning rate* $\eta \leq \frac{1}{16L(1+\log T)}$ *inside Eq.* (10) *of Alg. 1 guarantees,*

$$\mathbf{E}(f(x_T) - f(w^*)) = \mathbf{E}\Delta_T \leq \tilde{O}\left(\frac{D^2}{\eta T} + 2\eta\frac{\tilde{\sigma}^2}{T} + \frac{4D\tilde{\sigma}}{\sqrt{T}}\right) ,$$

*where* $\Delta_t := f(x_t) - f(w^*)$, *and* $\tilde{\sigma}^2 := 32D^2\sigma_L^2 + 2\sigma^2$.

And note that this demonstrates the same stability of this $\mu^2$-SGD variant, similarly to the stability of the variant that we discuss in the main text and in Thm. 4.2.

Next we provide a proof.

*Proof of Thm. G.2.* The proof is a direct combination of Thm. G.1 together with the standard regret bound of OGD (Online Gradient Descent), which in turn enables to utilize the Anytime guarantees of Thm. 3.1.

**Part 1: Regret Bound.** Standard regret analysis of the update rule in Eq. (9) implies the following for any $t$ (Hazan et al., 2016),

$$\sum_{\tau=1}^{t} \alpha_\tau d_\tau \cdot (w_\tau - w^*) \leq \frac{D^2}{2\eta} + \frac{\eta}{2}\sum_{\tau=1}^{t} \alpha_\tau^2 \|d_\tau\|^2 . \tag{33}$$

**Part 2: Anytime Guarantees.** Since the $x_t$'s are weighted averages of the $w_t$'s we may invoke Thm. 3.1, which ensures for any $t \in [T]$,

$$\alpha_{1:t}\Delta_t = \alpha_{1:t}(f(x_t) - f(w^*)) \leq \sum_{\tau=1}^{t} \alpha_\tau \nabla f(x_\tau) \cdot (w_\tau - w^*) ,$$

where we denote $\Delta_t := f(x_t) - f(w^*)$.

**Part 3: Combining Guarantees.** Combining the above Anytime guarantees together with the bound in Eq. (33) yields,

$$\alpha_{1:t}\Delta_t \leq \sum_{\tau=1}^{t}\alpha_\tau \nabla f(x_\tau)\cdot(w_\tau-w^*)$$

$$= \sum_{\tau=1}^{t}\alpha_\tau d_\tau\cdot(w_\tau-w^*) + \sum_{\tau=1}^{t}\alpha_\tau(\nabla f(x_\tau)-d_\tau)\cdot(w_\tau-w^*)$$

$$= \frac{D^2}{2\eta} + \frac{\eta}{2}\sum_{\tau=1}^{t}\alpha_\tau^2\|d_\tau\|^2 - \sum_{\tau=1}^{t}\alpha_\tau\epsilon_\tau\cdot(w_\tau-w^*)$$

$$\leq \frac{D^2}{2\eta} + \frac{\eta}{2}\sum_{\tau=1}^{t}\alpha_\tau^2\|\nabla f(x_\tau)+\epsilon_\tau\|^2 + \sum_{\tau=1}^{t}\|\alpha_\tau\epsilon_\tau\|\cdot\|w_\tau-w^*\|$$

$$\leq \frac{D^2}{2\eta} + \eta\sum_{\tau=1}^{t}\alpha_\tau^2\|\nabla f(x_\tau)\|^2 + \eta\sum_{\tau=1}^{t}\alpha_\tau^2\|\epsilon_\tau\|^2 + D\sum_{\tau=1}^{t}\|\alpha_\tau\epsilon_\tau\|$$

$$\leq \frac{D^2}{2\eta} + 2\eta L\sum_{\tau=1}^{t}\alpha_\tau^2\Delta_\tau + \eta\sum_{\tau=1}^{t}\alpha_\tau^2\|\epsilon_\tau\|^2 + D\sum_{\tau=1}^{t}\|\alpha_\tau\epsilon_\tau\|$$

$$\leq \frac{D^2}{2\eta} + 4\eta L\sum_{\tau=1}^{t}\Delta_\tau + \eta\sum_{\tau=1}^{t}\|\epsilon_\tau\|^2 + D\sum_{\tau=1}^{t}\|\epsilon_\tau\| , \tag{34}$$

where the first inequality follows from Cauchy-Schwartz; the second inequality holds since $\|w_t - w^*\| \leq D$, as well as from using $\|a+b\|^2 \leq 2\|a\|^2 + 2\|b\|^2$ which holds for any $a, b \in \mathbb{R}^d$, the third inequality follows by the self bounding property for smooth functions (see Lemma E.1) implying that $\|\nabla f(x_\tau)\|^2 \leq 2L(f(x_\tau) - f(w^*)) := 2L\Delta_\tau$; and the fourth inequality follows due to $\alpha_\tau = 1$.

Next, we will take expectation over Eq. (34), yielding,

$$t\mathbf{E}\Delta_t = \alpha_{1:t}\mathbf{E}\Delta_t \leq \frac{D^2}{2\eta} + 4\eta L\sum_{\tau=1}^{t}\mathbf{E}\Delta_\tau + \eta\sum_{\tau=1}^{t}\mathbf{E}\|\epsilon_\tau\|^2 + D\sum_{\tau=1}^{t}\mathbf{E}\|\epsilon_\tau\|$$

$$\leq \frac{D^2}{2\eta} + 4\eta L\sum_{\tau=1}^{t}\mathbf{E}\Delta_\tau + \eta\sum_{\tau=1}^{t}\mathbf{E}\|\epsilon_\tau\|^2 + D\sum_{\tau=1}^{t}\sqrt{\mathbf{E}\|\epsilon_\tau\|^2}$$

$$\leq \frac{D^2}{2\eta} + 4\eta L\sum_{\tau=1}^{t}\mathbf{E}\Delta_\tau + \eta\sum_{\tau=1}^{t}\tilde{\sigma}^2/\tau + D\sum_{\tau=1}^{t}\sqrt{\tilde{\sigma}^2/\tau}$$

$$\leq \frac{D^2}{2\eta} + 4\eta L\sum_{\tau=1}^{t}\mathbf{E}\Delta_\tau + \eta\tilde{\sigma}^2\sum_{\tau=1}^{t}1/\tau + D\tilde{\sigma}\sum_{\tau=1}^{t}\sqrt{1/\tau}$$

$$\leq \frac{D^2}{2\eta} + 4\eta L\sum_{\tau=1}^{t}\mathbf{E}\Delta_\tau + \eta\tilde{\sigma}^2(1+\log t) + 2D\tilde{\sigma}\sqrt{t}$$

$$\leq \frac{D^2}{2\eta} + \frac{1}{4(1+\log T)}\sum_{\tau=1}^{t}\mathbf{E}\Delta_\tau + \eta\tilde{\sigma}^2(1+\log t) + 2D\tilde{\sigma}\sqrt{t} , \tag{35}$$

where the second lines is due to Jensen's inequality implying that $\mathbf{E}X \leq \sqrt{\mathbf{E}X^2}$ for any random variable $X$; the third line follows from $\mathbf{E}\|\epsilon_t\|^2 \leq \tilde{\sigma}^2/t$ which holds by Thm. G.1. We also used $\sum_{\tau=1}^{t}1/\sqrt{\tau} \leq 2\sqrt{t}$ as well as $\sum_{\tau=1}^{t}1/\tau \leq 1+\log t$. Lastly, we use our choice for $\eta$.

To obtain the final bound we will apply Lemma G.3 below to Eq. (35).

**Lemma G.3.** *Let $T > 2$, and $\{A_t\}_{t\in[T]}$ be a sequence of non-negative elements and $\{\mathcal{B}_t \in \mathbb{R}\}_{t\in[T]}$ a monotonically increasing sequence of non-negative elements, and assume that for any $t \leq T$,*

$$tA_t \leq \mathcal{B}_t + \frac{1}{4(1+\log T)}\sum_{t=1}^{T}A_t ,$$

*Then the following bound holds $\forall t \in [T]$,*

$$A_t \leq 2\mathcal{B}_t/t .$$

Taking $A_t \leftarrow \mathbf{E}\Delta_t$ and $\mathcal{B}_t \leftarrow \frac{D^2}{2\eta} + \eta\tilde{\sigma}^2(1 + \log t) + 2D\tilde{\sigma}\sqrt{t}$ provides the following explicit bound,

$$\mathbf{E}(f(x_T) - f(w^*)) = \mathbf{E}\Delta_T \leq \tilde{O}\left(\frac{D^2}{\eta T} + 2\eta\frac{\tilde{\sigma}^2}{T} + \frac{4D\tilde{\sigma}}{\sqrt{T}}\right) ,$$

which concludes the proof. □

### G.1 PROOF OF LEMMA G.3

*Proof.* We shall prove the lemma by induction. For the base case we have,

$$A_1 \leq \mathcal{B}_1 + \frac{1}{4(1 + \log T)}A_1 \leq \mathcal{B}_1 + \frac{1}{4}A_1$$

This directly implies that $A_1 \leq 4\mathcal{B}_1/3 \leq 2\mathcal{B}_1$, which establishes the base case.

**For the induction step**, lets assume that the lemma holds for for any $\tau \leq t$, and show that it also holds for $t + 1$. Indeed, using the induction assumption we obtain,

$$(t+1)A_{t+1} \leq \mathcal{B}_{t+1} + \frac{1}{4(1 + \log T)}\left(A_{t+1} + \sum_{\tau=1}^{t} A_\tau\right)$$

$$\leq \mathcal{B}_{t+1} + \frac{1}{4}A_{t+1} + \frac{1}{4(1 + \log T)}\sum_{\tau=1}^{t}\frac{2\mathcal{B}_\tau}{\tau}$$

$$\leq \mathcal{B}_{t+1} + \frac{1}{4}A_{t+1} + \frac{2\mathcal{B}_{t+1}}{4(1 + \log T)}\sum_{\tau=1}^{t}\frac{1}{\tau}$$

$$\leq \mathcal{B}_{t+1} + \frac{1}{4}A_{t+1} + \frac{2\mathcal{B}_{t+1}}{4(1 + \log T)}\sum_{\tau=1}^{t}\frac{1}{\tau}$$

$$\leq \mathcal{B}_{t+1} + \frac{1}{4}A_{t+1} + \frac{1}{2}\mathcal{B}_{t+1} .$$

where we have used the monotonicity of the $\mathcal{B}_t$ sequence, as well as the fact that for any $t \leq T$ we have $\sum_{\tau=1}^{t}\frac{1}{\tau} \leq 1 + \log t \leq 1 + \log T$. Re-ordering the above implies $(t + 1 - \frac{1}{4})A_{t+1} \leq \frac{3}{2}\mathcal{B}_{t+1}$. Since $t \geq 1$ then $t + 1 - \frac{1}{4} \geq \frac{3}{4}(t + 1)$. Using this together with the non-negativity of $A_{t+1}$ directly implies that,

$$A_{t+1} \leq \frac{3\mathcal{B}_{t+1}/2}{t + 1 - \frac{1}{4}} \leq \frac{3\mathcal{B}_{t+1}/2}{3(t + 1)/4} = 2\mathcal{B}_{t+1} .$$

which establishes the induction step; and in turn the induction proof.

□

# H  EXPERIMENTS

## H.1  TECHNICAL DETAILS

We compared the following optimization algorithms over a range of fixed learning rates[4]:

- $\mu^2$-SGD.

- Momentum-based SGD with $\mu = 0.9$ and $\tau = 0.9$ (see PyTorch docs.[5]).

- Standard SGD.

- STORM.

- Anytime-SGD.

We evaluated our approach on the following datasets:

- **CIFAR-10 Dataset.** The CIFAR-10 dataset (Krizhevsky et al., 2014) consists of 60,000 color images across ten classes with a resolution of 32x32 pixels.

- **MNIST-10 Dataset.** The MNIST dataset (LeCun et al., 2010) comprises 70,000 grayscale images of handwritten digits (0-9) with a resolution of 28x28 pixels.

## H.2  CONVEX SETTING

In this experiment, we address a logistic regression problem on the MNIST dataset. Both the training and testing phases employed mini-batches of size 64, with one full pass (epoch) over the dataset.

Also, the model weights are constrained within a unit ball, limiting the solution space to a compact convex set:

$$\mathcal{K} = \{w \in \mathbb{R}^d : \|w\|_2 \leq 1\},$$

where $w \in \mathbb{R}^d$ denotes the model weights.

At each iteration, the weights are projected back into the unit ball using the projection function $\Pi_{\mathcal{K}}(w)$, ensuring that the norm of $w$ remains bounded.

The following algorithms were evaluated with their respective parameter settings: $\mu^2$-SGD with $\alpha_t = t$ and $\beta_t = 1/t$, STORM with $\beta_t = 1/t$, and Anytime-SGD with $\alpha_t = t$.

### H.2.1  TEST ACCURACY ACROSS LEARNING RATES

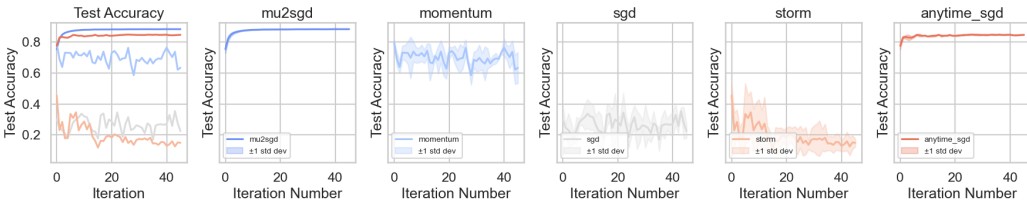

Figure 3: Test Accuracy at Learning Rate = 10 ($\uparrow$ is better).

---

[4]Note that comparing fixed learning rates across different optimizers is consistent with our theoretical findings, which demonstrate optimal convergence for $\eta = \alpha_t \eta_{\mu^2\text{-SGD}} = O(t/T) \simeq O(1)$.

[5]https://pytorch.org/docs/stable/generated/torch.optim.SGD.html

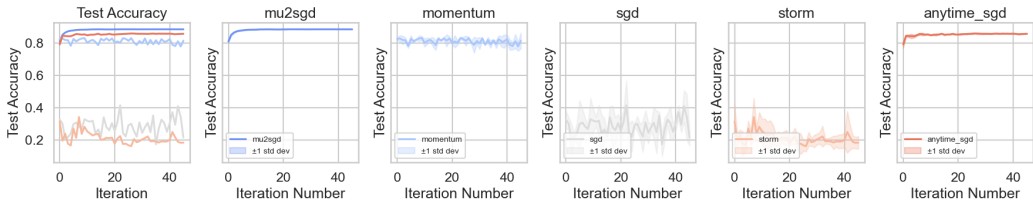

Figure 4: Test Accuracy at Learning Rate = 1 (↑ is better).

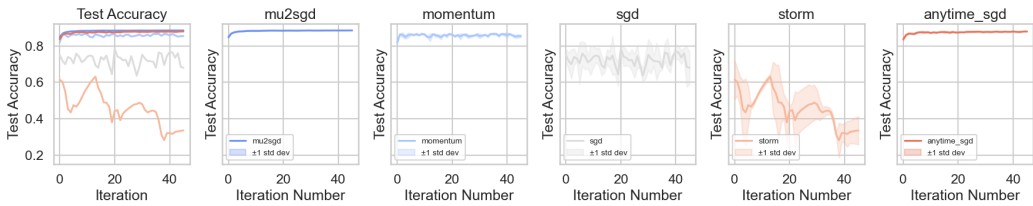

Figure 5: Test Accuracy at Learning Rate = 0.1 (↑ is better).

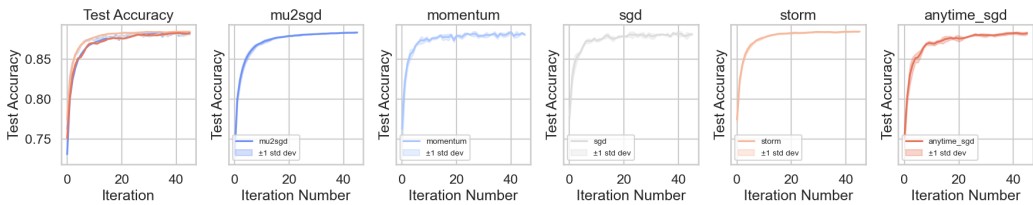

Figure 6: Test Accuracy at Learning Rate = 0.01 (↑ is better).

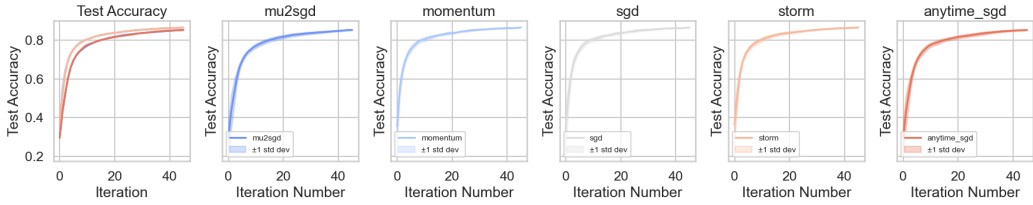

Figure 7: Test Accuracy at Learning Rate = 0.001 (↑ is better).

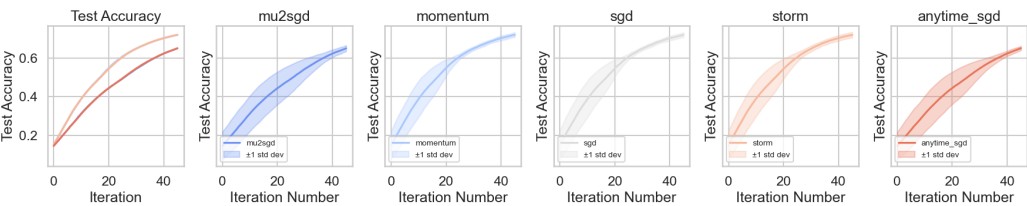

Figure 8: Test Accuracy at Learning Rate = 0.0001 (↑ is better).

### H.2.2 TEST LOSS ACROSS LEARNING RATES

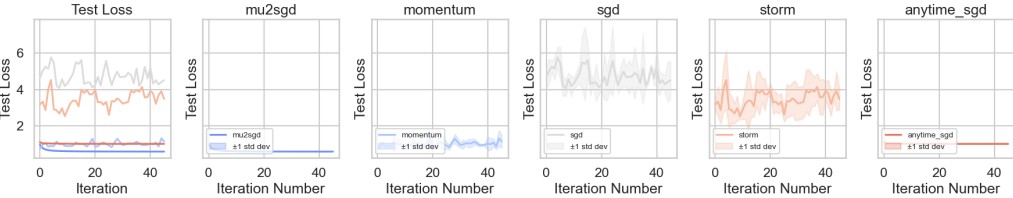

Figure 9: Test Loss at Learning Rate = 10 (↓ is better).

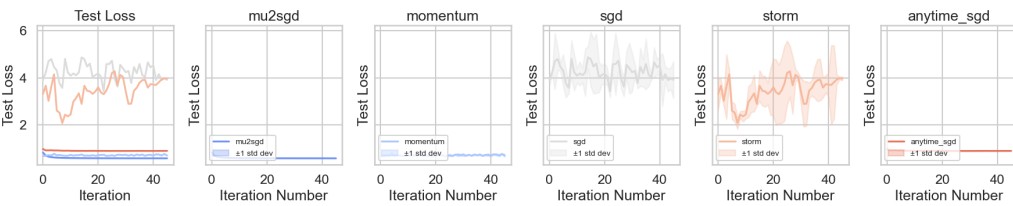

Figure 10: Test Loss at Learning Rate = 1 (↓ is better).

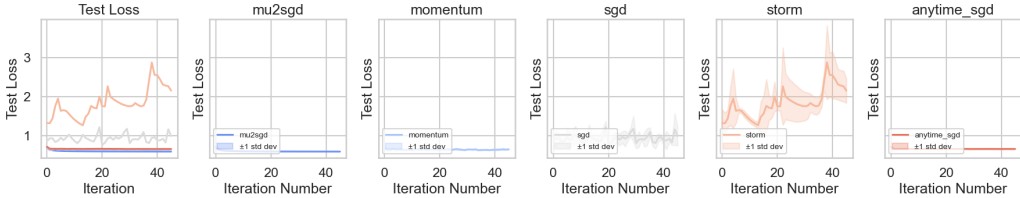

Figure 11: Test Loss at Learning Rate = 0.1 (↓ is better).

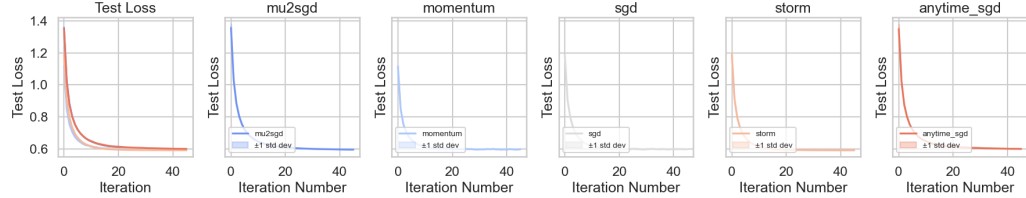

Figure 12: Test Loss at Learning Rate = 0.01 (↓ is better).

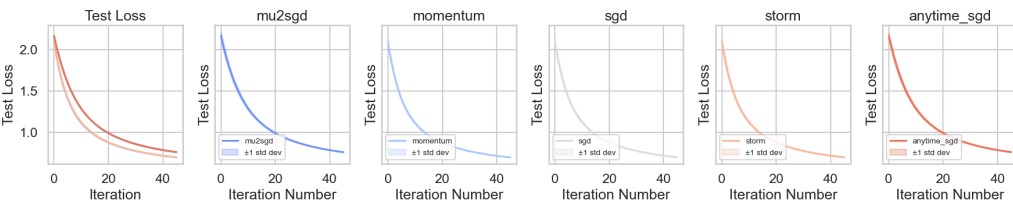

Figure 13: Test Loss at Learning Rate = 0.001 (↓ is better).

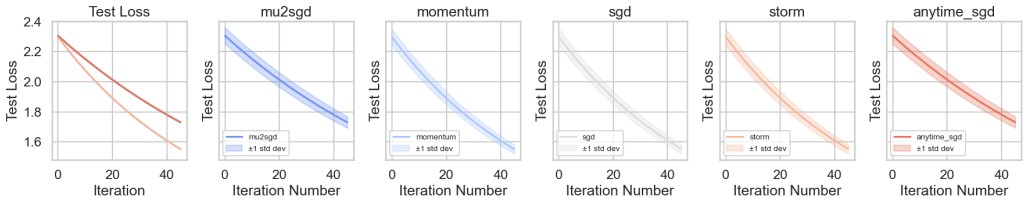

Figure 14: Test Loss at Learning Rate = 0.0001 (↓ is better).

## H.3 NON-CONVEX SETTING

**CIFAR-10 Experiment.** We trained ResNet-18 for 25 epochs using mini-batches of size 32. Training included RandomCrop (32×32, padding=2, p=0.5) and RandomHorizontalFlip (p=0.5) for data augmentation.

**MNIST-10 Experiment.** We trained a two-layer CNN with ReLU activations, max pooling, and two fully connected layers. Batch normalization was applied to the first fully connected layer. The model was trained using mini-batches of size 64 in a single pass over the dataset.

The following algorithms were evaluated with their respective *fixed* parameter settings: $\mu^2$-SGD with $\gamma_t = 0.1$ and $\beta_t = 0.9$, STORM with $\beta_t = 0.9$, and Anytime-SGD with $\gamma_t = 0.1$. These fixed parameter choices help mitigate the sensitivity of small momentum parameters in non-convex deep learning models, which arises from two key factors:

- Relevance of Historical Weights: In non-convex optimization, heavily relying on past weights (e.g., $\gamma_t \sim 1/t$) can hinder progress, as old weights may lose relevance or misguide optimization through saddle points and local minima.

- Numerical Stability: In high-parameter models or large datasets (where $t$ grows large), very small momentum parameters can introduce numerical instability, compromising optimization effectiveness and reliability.

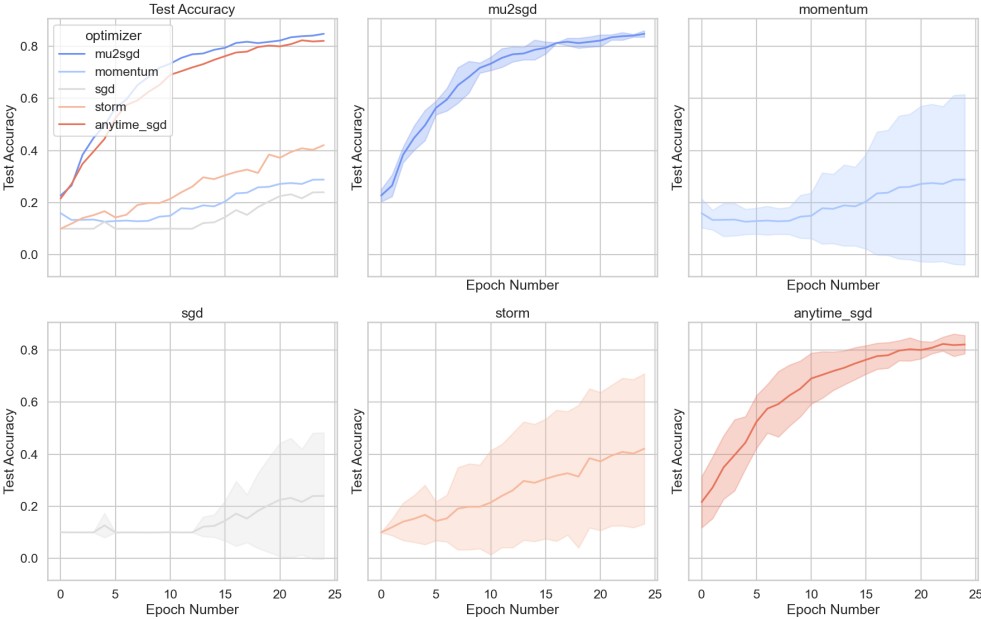

Figure 15: CIFAR-10: Test Accuracy Over Epochs at Learning Rate = 10 (↑ is better).

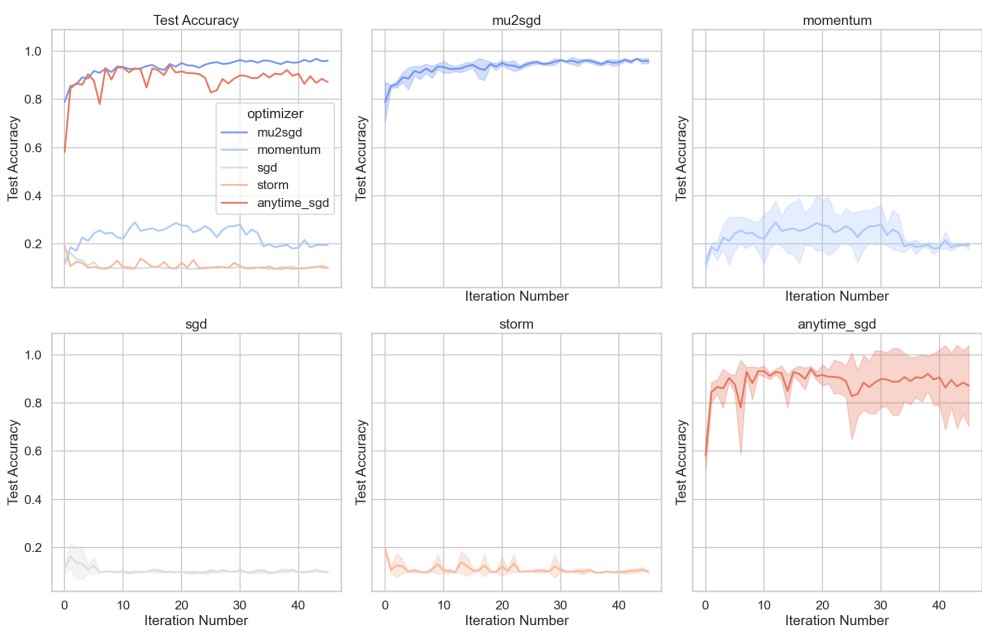

Figure 16: MNIST: Test Accuracy Over Iterations at Learning Rate=10 (↑ is better).

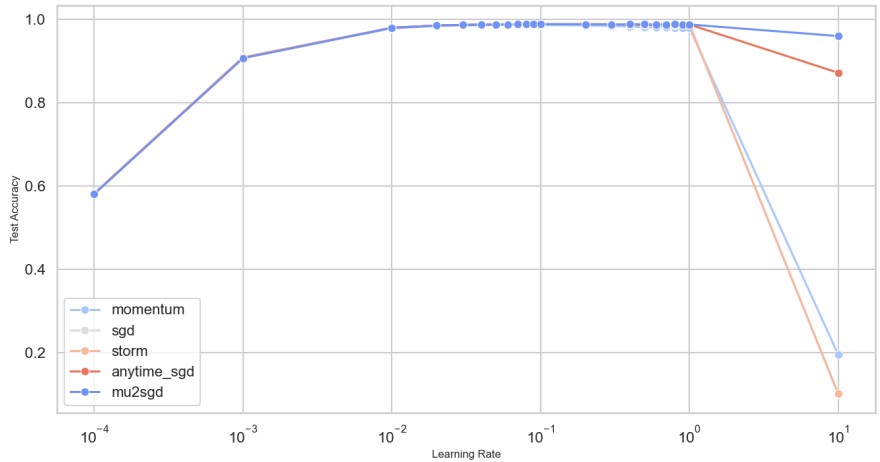

Figure 17: MNIST: Test Accuracy over LRs from 0.0001 to 10 (↑ is better).

### H.3.1 CIFAR-10: TEST ACCURACY ACROSS LEARNING RATES

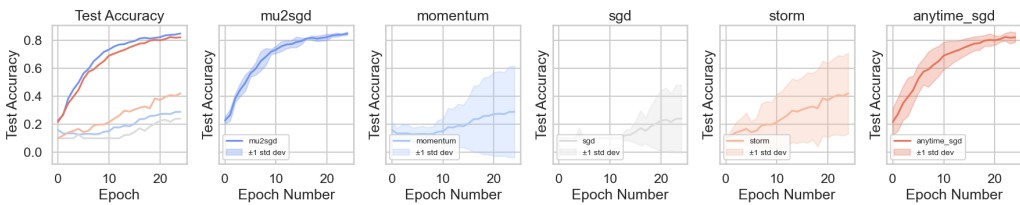

Figure 18: Test Accuracy at Learning Rate = 10 (↑ is better).

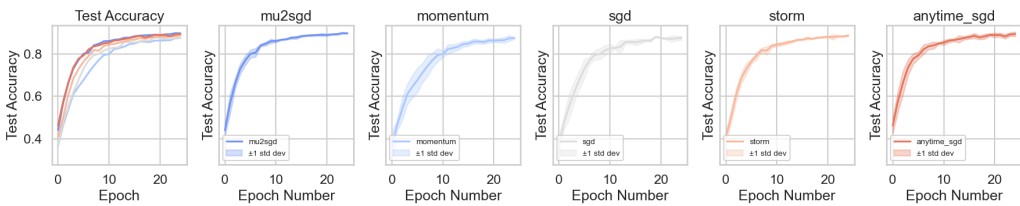

Figure 19: Test Accuracy at Learning Rate = 1 (↑ is better).

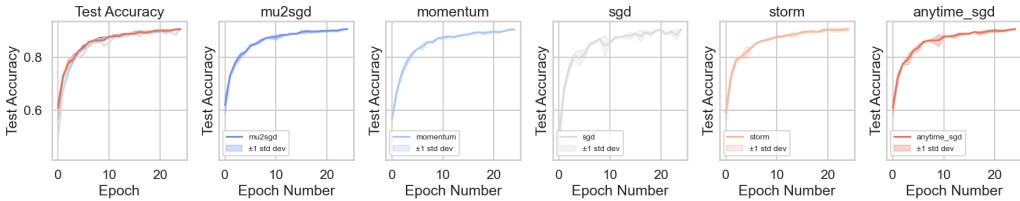

Figure 20: Test Accuracy at Learning Rate = 0.1 (↑ is better).

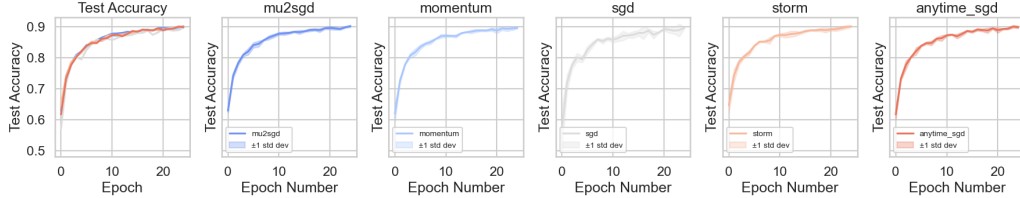

Figure 21: Test Accuracy at Learning Rate = 0.01 (↑ is better).

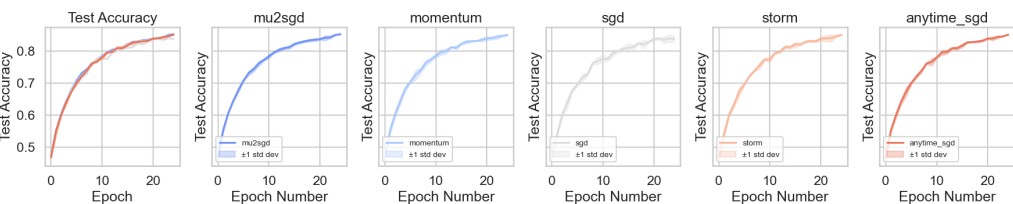

Figure 22: Test Accuracy at Learning Rate = 0.001 (↑ is better).

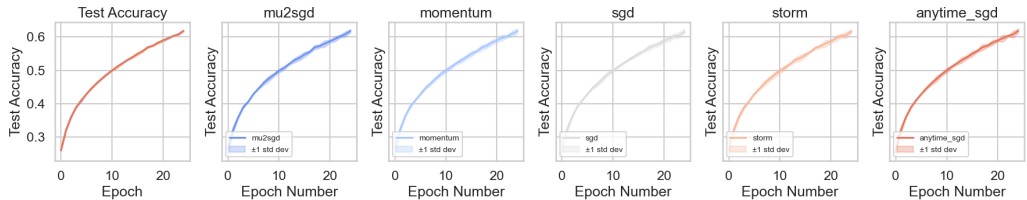

Figure 23: Test Accuracy at Learning Rate = 0.0001 (↑ is better).

### H.3.2 CIFAR-10: TEST LOSS ACROSS LEARNING RATES

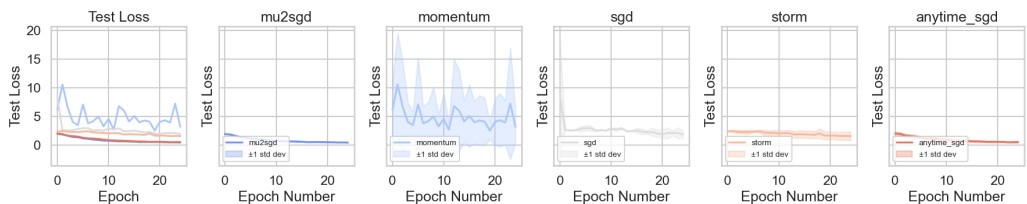

Figure 24: Test Loss at Learning Rate = 10 (↓ is better) [6].

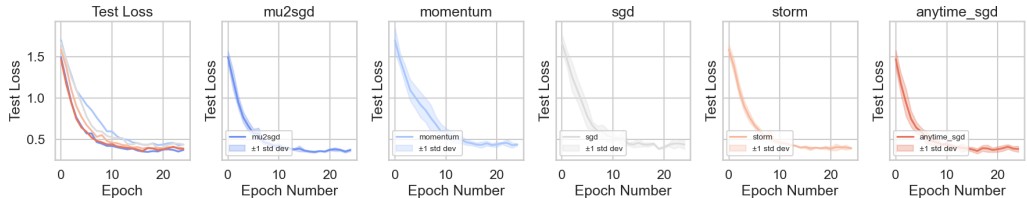

Figure 25: Test Loss at Learning Rate = 1 (↓ is better).

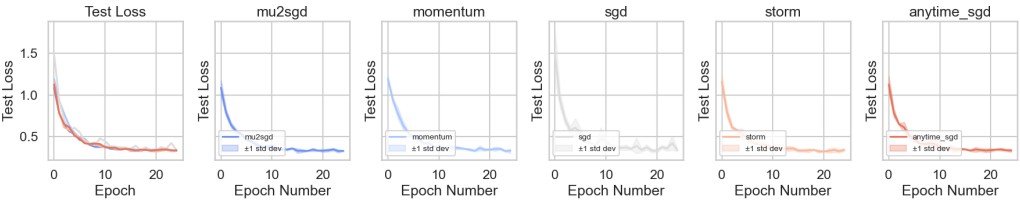

Figure 26: Test Loss at Learning Rate = 0.1 (↓ is better).

---

[6]Loss values are clipped at a maximum of 20 for better visualization.

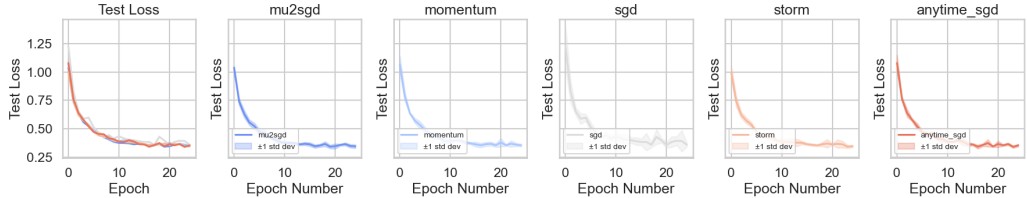

Figure 27: Test Loss at Learning Rate = 0.01 (↓ is better).

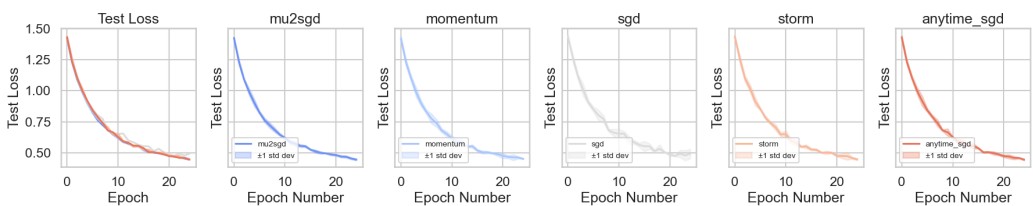

Figure 28: Test Loss at Learning Rate = 0.001 (↓ is better).

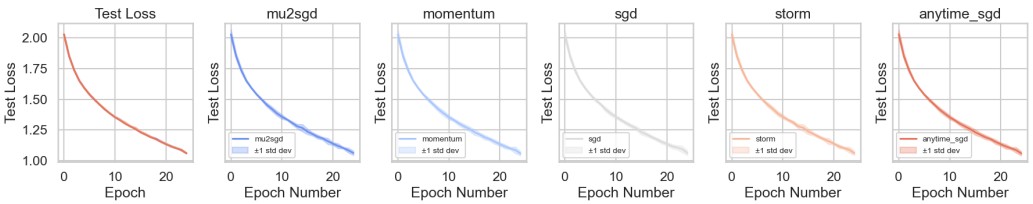

Figure 29: Test Loss at Learning Rate = 0.0001 (↓ is better).

### H.3.3 MNIST: TEST ACCURACY ACROSS LEARNING RATES

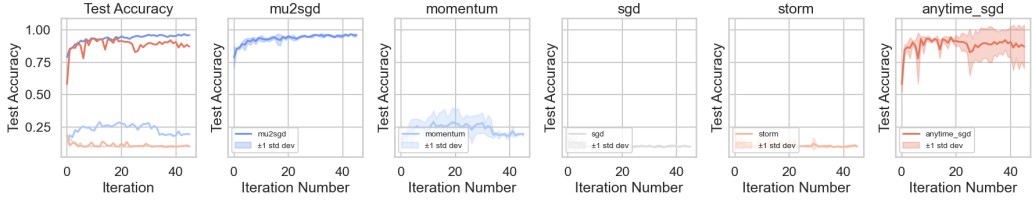

Figure 30: Test Accuracy at Learning Rate = 10 (↑ is better).

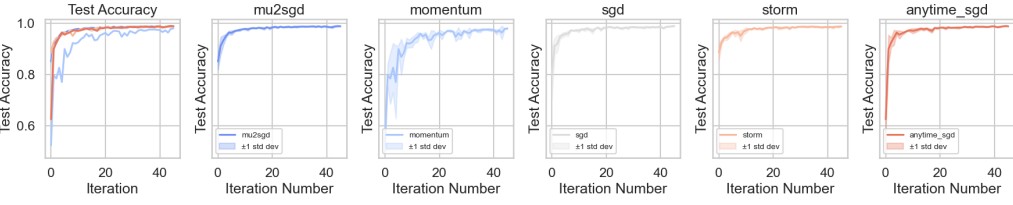

Figure 31: Test Accuracy at Learning Rate = 1 (↑ is better).

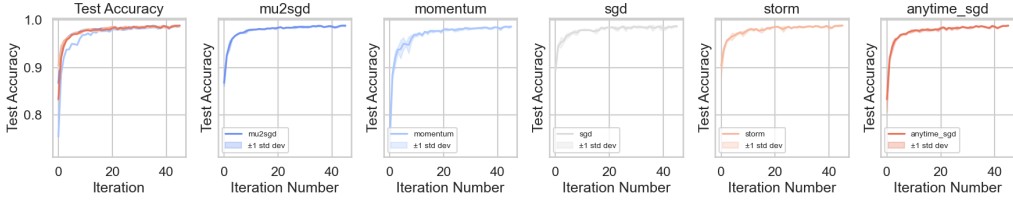

Figure 32: Test Accuracy at Learning Rate = 0.1 (↑ is better).

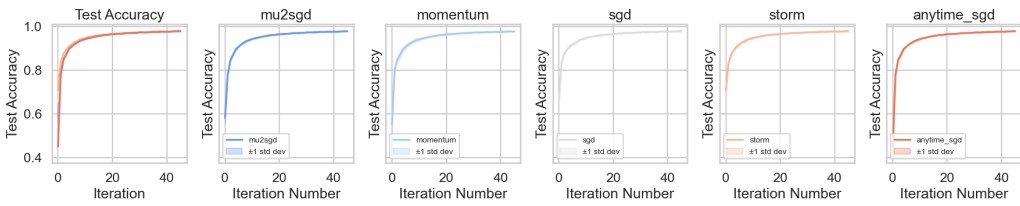

Figure 33: Test Accuracy at Learning Rate = 0.01 (↑ is better).

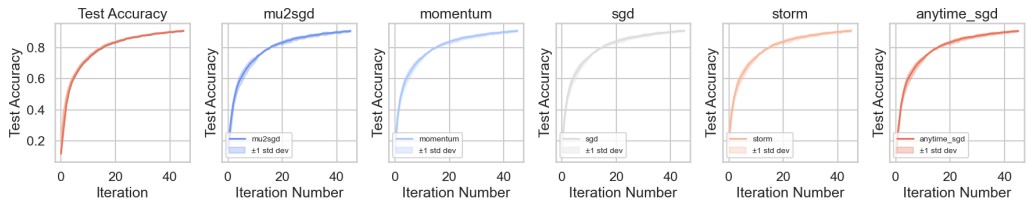

Figure 34: Test Accuracy at Learning Rate = 0.001 (↑ is better).

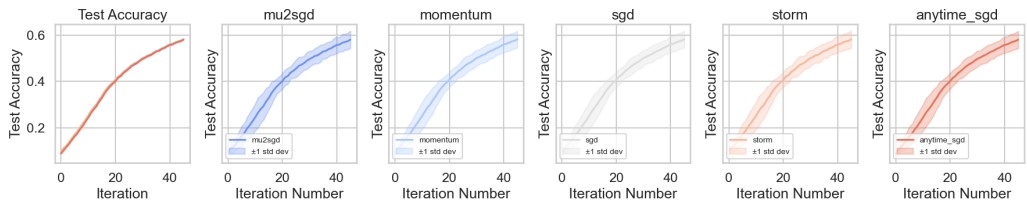

Figure 35: Test Accuracy at Learning Rate = 0.0001 (↑ is better).

### H.3.4   MNIST: TEST LOSS ACROSS LEARNING RATES

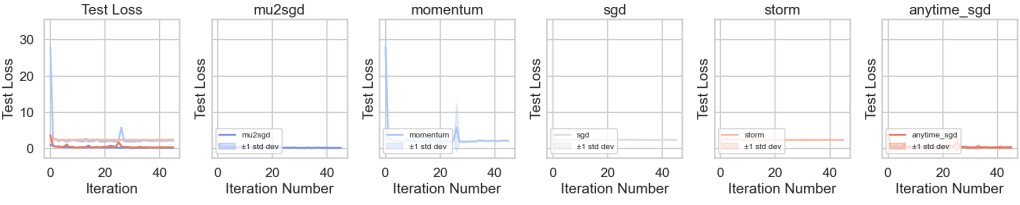

Figure 36: Test Loss at Learning Rate = 10 (↓ is better).

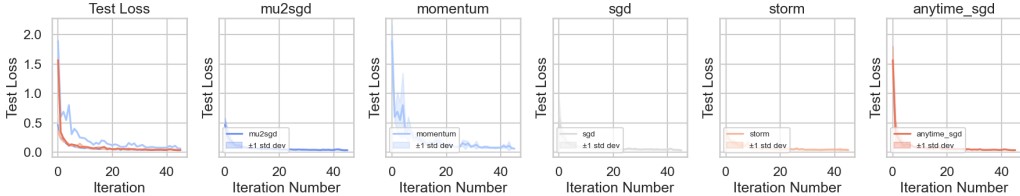

Figure 37: Test Loss at Learning Rate = 1 (↓ is better).

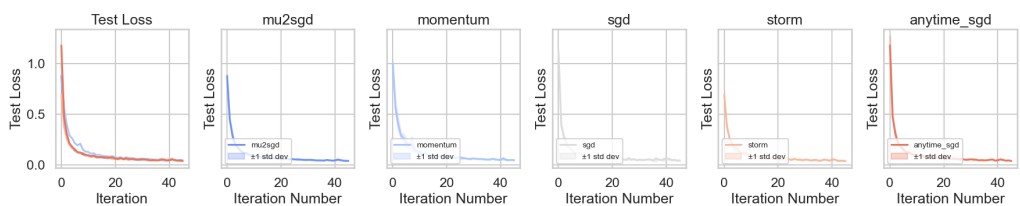

Figure 38: Test Loss at Learning Rate = 0.1 (↓ is better).

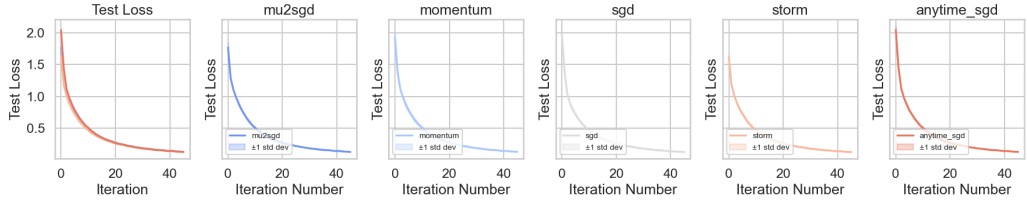

Figure 39: Test Loss at Learning Rate = 0.01 (↓ is better).

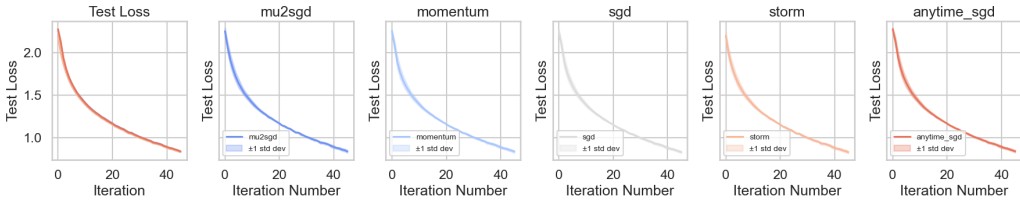

Figure 40: Test Loss at Learning Rate = 0.001 (↓ is better).

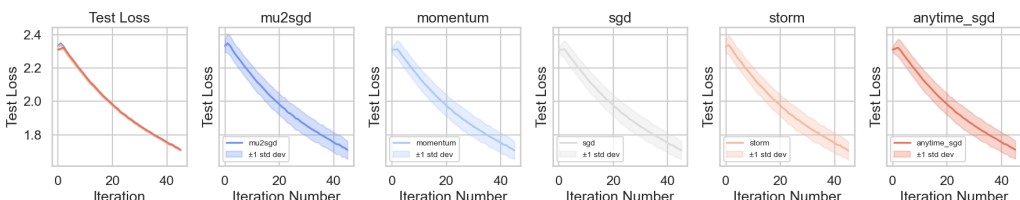

Figure 41: Test Loss at Learning Rate = 0.0001 (↓ is better).

