# OpenReview forum: "Do Stochastic, Feel Noiseless: Stable Stochastic Optimization via a Double Momentum Mechanism"
_ICLR.cc/2025/Conference — ICLR 2025 Poster_

### Official Review · Reviewer_pANq · 2024-11-01

**Soundness:** 3
**Presentation:** 3
**Contribution:** 2
**Rating:** 6
**Confidence:** 4

**Summary:**

This paper proposes a novel gradient estimator for stochastic convex optimization, combining momentum-based techniques. Using this estimator, the authors develop robust SGD-style algorithms that achieve optimal convergence rates in both noiseless and noisy settings, maintaining stable performance over a wide range of learning rates.

**Strengths:**

Considering algorithms that work in both noisy and noise-free conditions helps improve the robustness of the algorithm.

**Weaknesses:**

1.	The structure of this paper makes it difficult to follow, especially in terms of understanding the novelty of the algorithms.
2.	The experimental results presented do not convincingly demonstrate the superiority of the proposed algorithm. The choice of learning rates between 10 and 1000 is unconventional and the hyperparameter settings for STORM are not clearly defined. Furthermore, the generalization benefits mentioned by the author in line 52 has no experimental support.
3.	The novelty of this paper appears to be limited. For example, STORM was originally designed for a non-convex setting, and one of the main contributions of this paper seems to be the redesign of STORM's parameters for a convex setting.
4.	Many fully parameter-free algorithms[1,2] have appeared recently, which do not require knowledge of the smoothing constant $L$ and can achieve the same convergence rate. In contrast, this paper still relies on smoothing constants to determine the learning rate. Under fully parameter-free conditions, even a wide range of learning rate options appears to lose its significance.

[1] Khaled A, Jin C. Tuning-Free Stochastic Optimization[J]. arXiv preprint arXiv:2402.07793, 2024.
[2] Ivgi M, Hinder O, Carmon Y. Dog is sgd’s best friend: A parameter-free dynamic step size schedule[C]//International Conference on Machine Learning. PMLR, 2023: 14465-14499.

**Questions:**

1.	Could the author provide more comparisons between the proposed algorithm and parameter-free algorithms?
2.	Could the author include additional experiments on the impact of noise on the algorithm?

---

> ### Author Response · Authors · 2024-11-21
>
> - Regarding the *Choice of Learning Rates*:
>
> Thank you for highlighting this important point. In the revised version of the rebuttal, we have included a close-up view in Figures 1 and 2, focusing on the performance within *conventional ranges* of learning rates for each setup for greater clarity. These close-ups highlight the remarkable stability of $\mu^2$-SGD, which is particularly valuable for achieving strong performance across a wide range of learning rates, including commonly used values. While $\mu^2$-SGD may not always provide the absolute best result, its consistently high performance and robustness greatly reduce the need for an expensive and exhaustive search for a high-performing learning rate. Additionally, learning rate ranges can vary between setups and can be both narrow and small in scale, which makes identifying the optimal range particularly challenging. $\mu^2$-SGD's ability to perform well across higher and broader learning rate ranges alleviates this difficulty, making it a highly flexible and reliable choice for a variety of setups.
>
> - Regarding the *Hyperparameter Settings for STORM*:
>
> Thank you for pointing that out. In our empirical evaluation, we implemented the version of STORM as presented in our work to evaluate its performance independently, without the influence of the AnyTime component. This approach was designed to clearly illustrate the benefits of integrating both components into the $\mu^2$-SGD algorithm. For consistency, we applied the same hyperparameter settings for STORM as those used in the $\mu^2$-SGD algorithm. This ensured a fair comparison and highlighted how the combined algorithm achieves superior performance and robustness. While each component may demonstrate suboptimal performance in isolation, their integration forms a powerful optimizer that consistently yields high performance and stability across a wide range of learning rates.
>
> - Regarding the *Experimental Support for Line 52*:
>
> It seems there was a misunderstanding regarding line 52. The generalization benefits we referenced are supported by the experiments included in the paper. Specifically, our results illustrate the convergence and robustness of the algorithm across a wide range of learning rates, providing empirical evidence for its stability and reliable performance. These findings directly align with the theoretical guarantees discussed.

---

> > ### Author Response · Authors · 2024-11-21
> >
> > - Regarding the *Novelty of Our Approach*:
> >
> > We appreciate your feedback and the opportunity to clarify the contributions of our work. The novelty of our approach extends well beyond adapting STORM’s parameters for a convex setting. Specifically, our method achieves several novel guarantees that distinguish it from prior work:
> >
> > 1. **Substantial Reduction in Variance:**  Our approach substantially reduces stochastic noise proportional to the (current) total number of iterations, leading to enhanced stability.
> >
> > 2. **Robustness Across a Wide Range of Learning Rates:**  Unlike many existing methods, our algorithm demonstrates strong performance and stability across a wide range of learning rates, reducing sensitivity to this critical hyperparameter.
> >
> > 3. **Eliminating Hyperparameter Tuning (Convex Setting):**  By leveraging the predefined momentum parameters derived from our theoretical framework, our approach achieves superior robustness and high performance across a wide range of learning rates, *completely eliminating* the need for hyperparameter tuning of both momentums and the learning rate (see Figure 1). This prevents costly hyperparameter searches commonly required in traditional optimization methods. This not only streamlines the optimization process but also maintains strong and robust empirical performance.
> >
> > 4. **Non-Trivial Algorithmic Design and Analysis:** On the technical side, it is important to note that individually, each of the momentum techniques that we combine is unable to ensure the stability properties that we are able to ensure for their appropriate combination, i.e. for $\mu^2$-SGD and for $\mu^2$-ExtraSGD. Moreover, our accelerated version $\mu^2$-ExtraSGD, requires a careful and delicate blend of several techniques in the right interweaved manner, which leads to a concise yet delicate analysis. The novelty here is with respect to both the highly non-trivial algorithmic design and with respect to the delicate (yet concise) analysis.
> >
> > We hope this explanation provides greater clarity regarding the unique aspects of our approach. Thank you for raising this point.
> >
> > - Regarding the *Comparison to Parameter-Free Methods*:
> >
> > Existing parameter-free and adaptive methods focus on adapting the learning-rate schedule while using the standard gradient estimates, similarly to SGD. Thus, a smart learning rate adaptation is the key to their stability.
> > Conversely, in our approach, we achieve stability by designing and employing a new gradient estimate without learning rate adaptation. Therefore, our approach is orthogonal to existing parameter-free approaches and in a sense complements them. Therefore, we do not think that our approach should be compared with these parameter-free approaches. In fact, since the sources of stability are complementary, an interesting future direction that we intend to explore is combining our approach with existing parameter-free approaches, and we believe that this has the potential to substantially boost performance.

---

> ### Comment · Reviewer_pANq · 2024-11-22
> **Comments**
>
> Thank you for your rebuttal. I will increase my score to 6.

---

### Official Review · Reviewer_eNgP · 2024-11-03

**Soundness:** 3
**Presentation:** 2
**Contribution:** 2
**Rating:** 6
**Confidence:** 3

**Summary:**

Combining two existing methods, the authors propose an accelerated SGD mechanism that is stable w.r.t. learning rate. The authors provide rigorous justifications for their claims and supporting experiments.

**Strengths:**

1. By combining two existing methods--Anytime-SGD (Cutkosky, 2019) and STORM (STochastic Recursive Momentum) (Cutkosky & Orabona, 2019), the authors propose $\mu^2$-SGD.

2. The authors show the error term is upper bounded by $\frac{1}{t}$ with $\mu^2$-SGD. Moreover, they obtain an upper bound for the excess loss that gives a noise independent optimal learning rate, which results in a wider range of optimal choice for learning rate compared to SGD.

3. By adding Optimistic-OGD, the authors propse an accelerated version of $\mu^2$-SGD--$\mu^2$-ExtraSGD. The optimal learning rate in this case is constant in time so the optimal rate can be obtained without doing time-varying learning rate.

4. The authors have experiments in both convex and non-convex setting to verify their results. Experiments show $\mu^2$-SGD and $\mu^2$-ExtraSGD are stable w.r.t different learning rates while other methods don't. Experiments are done thoroughly with details given in the appendix.

**Weaknesses:**

1. The main contribution of this work--$\mu^2$-SGD and $\mu^2$-ExtraSGD--comes by combining two existing works. The proofs for main theorems are quite traditional. One could argue this lacks novelty, although I personally found the results interesting.

2. In the numerical experiments (non-convex setting), It seems $\mu^2$-SGD and $\mu^2$-ExtraSGD only make a difference when the learning rate is far away from normal choices. More precisely, Figure 2 shows all methods are quite similar for $\eta\leq1$. $\mu^2$-SGD and $\mu^2$-ExtraSGD are better only when $\eta >1$, which is not a typical choice for learning rate anyway. I am not sure how learning rate stability is appreciated by the community.

3. By allowing a wider range of learning rates, one could argue that with this method, we don’t need to tune the learning rate anymore, like we could set it to be 1. However, $\mu^2$-SGD and $\mu^2$-ExtraSGD are not hyperparameter free, i.e. they introduce weights $\alpha_t$ and Corrected Momentum weights $\beta_t$, which are chosen to be $t$ and $\frac{1}{t}$ in the experiments. I wish the authors could give more insights into this topic. If the cost of reducing one hyperparameter is introducing two hyperparameters, I am not sure it’s worth it.

4. The experiments are done for different learning rates, but I found the plots were hard to read when the accuracies/losses got too close. Presenting results with an additional table could help in these cases.

**Questions:**

See weaknesses.

---

> ### Author Response · Authors · 2024-11-21
>
> Thank you for your valuable feedback. We wish to address some points raised to provide further clarification and insight into our work:
>
> - Regarding the *Novelty of Our Theory*:
>
> On the conceptual level, we indeed integrate elements from existing techniques, which is in fact, one of the strengths of our approach. By building on these foundations, we achieve *novel guarantees*, including (1) a substantial reduction in variance proportional to the (current) total number of iterations and (2) robustness across a wide range of learning rates.
>
> On the technical side, it is important to note that individually, each of the momentum techniques that we combine is unable to ensure the stability properties that we are able to ensure for their appropriate combination, i.e., for $\mu^2$-SGD and $\mu^2$-ExtraSGD. Moreover, our accelerated version $\mu^2$-ExtraSGD, requires a careful and delicate blend of several techniques in the right interweaved manner, which leads to a concise yet delicate analysis. The novelty here is with respect to both the highly non-trivial algorithmic design and with respect to the delicate and concise analysis.
>
> - Regarding the *Choice of Learning Rates*:
>
> Thank you for highlighting this important point. In the revised version of the rebuttal, we have included a close-up view in Figures 1 and 2, focusing on the performance within *typical ranges* of learning rates for each setup for greater clarity. These close-ups highlight the remarkable stability of $\mu^2$-SGD, which is particularly valuable for achieving strong performance across a wide range of learning rates, including commonly used values. While $\mu^2$-SGD may not always provide the absolute best result, its consistently high performance and robustness greatly reduce the need for an expensive and exhaustive search for a high-performing learning rate. Additionally, learning rate ranges can vary between setups and can be both narrow and small in scale, which makes identifying the optimal range particularly challenging. $\mu^2$-SGD's ability to perform well across higher and broader learning rate ranges alleviates this difficulty, making it a highly flexible and reliable choice for a variety of setups.
>
> - Regarding the *Momentum Parameters*:
>
> We would like to clarify that the parameters $\alpha_t = t$ and $\beta_t = 1/t$ in the convex setup are not free hyperparameters but are deterministic values that depend solely on the iteration number. These parameters are derived directly from our theoretical analysis and designed to ensure stability and convergence. By using these predefined parameters in practice, our approach achieves superior robustness and high performance across a wide range of learning rates, *completely eliminating* the need for hyperparameter tuning of both momentums and the learning rate (see Figure 1). This prevents costly hyperparameter searches commonly required in traditional optimization methods. We believe this aspect highlights a significant practical advantage of our approach, as it reduces the computational overhead associated with hyperparameter tuning while maintaining strong empirical performance and robustness.
>
> - Regarding the *Presentation of Results*:
>
> The revised version of the rebuttal has been updated (main paper and appendix) to include additional close-up views of the figures and separate figures for each optimizer and learning rate with shared axes to ensure clarity and facilitate comparison.

---

> > ### Comment · Reviewer_eNgP · 2024-11-22
> > **Response to authors**
> >
> > Thank you for your rebuttal. The new figures do help me to understand the advantages of the proposed algorithms. While I still have some concerns about the unusual choices of learning rate, I believe some people can benefit from learning rate stability. I have increased my rating to 6. Good luck!

---

### Official Review · Reviewer_p8j2 · 2024-11-03

**Soundness:** 3
**Presentation:** 3
**Contribution:** 3
**Rating:** 6
**Confidence:** 3

**Summary:**

The paper introduces the $\mu^2$ (Momentum$^2$) gradient estimator, designed to manage gradient noise in stochastic optimization, particularly in convex and smooth loss scenarios using single-sample batch sizes. This estimator integrates two momentum techniques:

1. **Anytime Averaging**, which averages the query points for the gradient oracle.
2. **STORM**, a corrected momentum method that averages gradient estimates with bias correction.

By combining these techniques, $\mu^2$ achieves a progressively shrinking square error of the gradient estimates ($\|\epsilon_t\|^2 \propto 1/t$), contrasting with the fixed gradient error in standard stochastic optimization ($\|\epsilon_t\|^2 = O(1)$). This property enables the use of a fixed step-size for convergence, removing the necessity of step-size decay in stochastic settings. Additionally, $\mu^2$ allows the norm of the gradient estimate to serve as a stopping criterion.

The paper implements $\mu^2$ in two algorithms:

1. **$\mu^2$-SGD**: This combines $\mu^2$ with SGD. Although $\mu^2$ could work with other first-order methods, the authors focus on SGD to derive theoretical guarantees. Theorem 4.2 shows that $\mu^2$-SGD achieves optimal convergence rates for noiseless ($O(L/T)$) and noisy ($O(L/T + \tilde{\sigma}/\sqrt{T})$) settings using a fixed learning rate $\eta_{\text{Offline}} = 1/8LT$. Remarkably, this rate does not need to change between noiseless and noisy conditions.

2. **$\mu^2$-ExtraSGD**: This accelerated variant of $\mu^2$-SGD uses the ExtraGradient framework to achieve optimal convergence rates of $O(L/T^2)$ (noiseless) and $O(L/T^2 + \tilde{\sigma}/\sqrt{T})$ (noisy) with a fixed learning rate $\eta_{\text{Offline}} = 1/2L$.

Experiments on MNIST and CIFAR-10 datasets, using convex (logistic regression) and non-convex (deep learning) models, demonstrate $\mu^2$-SGD's stability and performance over a range of learning rates compared to various baseline optimizers, including SGD, Polyak momentum, and individual applications of Anytime Averaging and STORM.

**Strengths:**

- The paper is well-organized, with clear explanations of each momentum component and its integration into $\mu^2$. A convergence guarantee for stochastic optimization with a fixed step-size addresses a significant challenge in optimization.

- The theoretical results are rigorously supported. Extending Theorem 4.2 to the accelerated version in Theorem 5.1 provides a pleasing completeness to the theory. In the former, the learning rate has a dependency on $T$, which prevents it from achieving the accelerated rate. This dependency is eliminated by introduction of the $\mu^2$-ExtraSGD.

- The theoretical results carry over nicely in the convex experiment, logistic regression on MNIST. $\mu^2$SGD consistently achieves the best performance across a very wide range of learning rates ($10^{-3}$ to $10^3$). Unfortunately, I am unable to tell where each method lies in the last plot in Figure 1 ($10^{-4}$).

**Weaknesses:**

- My main critique of this paper goes back to its main promise: with $\mu^2$SGD, convergence using a fixed step-size is achievable in a stochastic setting. However, although the step-size is fixed, varying momentum parameters, $\alpha$ and $\beta$, are required to prove convergence in Theorem 4.2. The decreasing step-size nature necessary for convergence in a stochastic setting seems to be delegated to the momentum parameter here. That being said, I appreciate the theoretical results and the simplicity of the schedule for $\alpha$. This same schedule was also used in the experiments, and it seems to work well.
- There is no experiment on $\mu^2$-ExtraSGD. Observing the accelerated rate in experiments could have potentially highlighted the strength of this method. I would also be curious to see how this acceleration performs compared to the Nesterov accelerated method.
- Although the paper shows $\mu^2$’s stability across learning rates, there is little analysis on how sensitive the algorithm might be to momentum parameter choices, especially in non-convex settings where fixed parameters are used.

**Questions:**

I am curious to know about your choice of  single sample per iterate setting. I understand that this choice might be more convenient to work in theory, but could you still carry the same theory using batch size b?

---

> ### Author Response · Authors · 2024-11-21
>
> Thank you for your valuable feedback. We wish to address some points raised to provide further clarification and insight into our work:
>
> - Regarding the *Presentation of Results*:
>
> The revised version of the rebuttal has been updated (main paper and appendix) to include additional close-up views of the figures and separate figures for each optimizer and learning rate with shared axes to ensure clarity and facilitate comparison.
>
> - Regarding the *Decaying Momentum Parameters*:
>
> Thank you for highlighting this point. The decaying momentum parameters $\beta_t = \frac{1}{t}$ and $\gamma_t\sim\frac{1}{t}$ are indeed necessary to ensure the theoretical guarantees of our framework. Nevertheless, a notable advantage of these parameters is that they are predefined and require no tuning, which, combined with our method's robustness across a wide range of learning rates, *eliminates the need for costly hyperparameter tuning*. This simplicity makes our approach practical and easy to apply, as demonstrated in our experiments for the convex setup (see Figure 1).
>
> - Regarding the *Addional Expriments for μ2-ExtraSGD*:
>
> We agree that demonstrating the accelerated performance of $\mu^2$-ExtraSGD through experiments could further support its theoretical guarantees. We will make an effort to include such experiments in the final version of the paper to better highlight its advantages.
>
> - Regarding the *Sensitivity of the Algorithm to Momentum Parameter Choices*:
>
> Thank you for raising this important point. The sensitivity to momentum parameter choices can arise from two main reasons:
>
> 1. **Relevance of Historical Weights in Non-Convex Problems**: In non-convex optimization, relying heavily on historical weights (e.g., using $\gamma_t \sim 1/t$ for the AnyTime momentum) can pose challenges since the nature of non-convex problems involves navigating through saddle points and local minima.  In such cases, old weights may not only lose relevance but also compromise progress by misguiding the optimization process.
>
> 2. **Numerical Stability in Complex Problems**: In high-parameter models or when working with large datasets (then $t$ becomes very large), the use of very small momentum parameters may introduce numerical instability, potentially compromising the effectiveness and reliability of the optimization process.
>
> To address this concern, we will add this discussion in the final version of our paper.
>
> - Regarding the *Extension of the Theory to Minibatches*:
>
> The theory can indeed be extended to incorporate a batch size $b > 1$. Consistent with minibatch theory [1], this extension leads to a comparable reduction in noise variance, where, in our case, $\||\varepsilon_t\||^2 \leq O(1/tb)$. Extending the framework to minibatches is both feasible and well-aligned with established results in minibatch stochastic optimization literature.
>
> [1] Dekel, Ofer, et al. "Optimal Distributed Online Prediction Using Mini-Batches." Journal of Machine Learning Research 13.1 (2012).

---

> > ### Comment · Reviewer_p8j2 · 2024-11-22
> > **Reply to author response**
> >
> > Thank you for your response. I will maintain my positive score and recommend accepting this paper.

---

### Official Review · Reviewer_8uRS · 2024-11-11

**Soundness:** 3
**Presentation:** 3
**Contribution:** 3
**Rating:** 6
**Confidence:** 4

**Summary:**

This paper presents two methods for stochastic optimization that achieve similar guarantees as standard gradient descent and accelerated gradient descent while being robust to the choice of learning rate involved.

**Strengths:**

This is a well written paper that presents two interesting algorithms with strong theoretical guarantees. The algorithms definitely appear novel to my knowledge but I do not know of the latest developments in this area.

**Weaknesses:**

The empirical section seems to be fairly underbaked in my opinion.

**Questions:**

1. Have the authors tested out the proposed algorithms (or their adaptive gradient variants) on problems that appropriately reflect problems/setups that are of realistic practical interest? For instance, training transformer based architectures on standard pre-training tasks?
2. Can the authors clarify how this algorithm's guarantees look like with (a) strong convexity, (b) non-convex but say with a PL style condition?

---

> ### Author Response · Authors · 2024-11-21
>
> Thank you for your valuable feedback. We wish to address some points raised to provide further clarification and insight into our work:
>
> - Regarding the *Empirical Evaluation*:
>
> To ensure transparency, we included the full implementation of our algorithms and experimental setup in the supplemental materials as part of the original submission. We have now also added our analysis notebooks and more detailed documentation for further clarity. We would be happy to address any additional questions or concerns as needed.
> Moreover, we agree that testing the proposed algorithms on more complex setups, such as transformer-based architectures, as well as exploring their adaptive variants, are interesting and important directions that merit further investigation. Since the primary objective of this paper is to introduce a novel concept and its potential impact, we have identified these aspects as a part of our planned future work.
>
>
> - Regarding the *Guarantees in other scenarios*:
>
> Thank you for raising this important point. Finding and analyzing an appropriate variant of our approach for the strongly convex case is non-trivial, but we believe that it can be done. What we believe that can be achieved is a result that demonstrates a convergence rate of $O(\frac{L/H}{T^3}+ \frac{\sigma D}{H T}$ for the case of $H$ strongly-convex functions, as well as demonstrate learning rate stability for $\eta \in [ \eta_\min,\eta_\max]$ where $\eta_\min = \frac{1}{Ht}$ and $\eta_\max = 1/L$. This is highly non-trivial, and we intend to investigate this question in the near future.
> The non-convex case is even less trivial since the AnyTime mechanism does not apply to such problems. Not even for functions that satisfy the PL condition. It will be interesting to understand what can be done in the non-convex case, and investigating with PL condition will be an excellent starting point.

---

> > ### Comment · Reviewer_8uRS · 2024-11-21
> > **Re. Author Response**
> >
> > Thanks for the clarifications. With regards to the strongly convex case, the 1/T^3 rate on the initial error seems to be rather a sloppy upper bound, at least in my experience. There is quite likely an easy way to achieve exponential rate of decay on the initial error (with error halving every condition number of \sqrt{condition number} steps) while continuing to achieve near optimal rates on the variance term (upto say a factor of two or so) by using some kind of an epoch based restart algorithm. Nevertheless, thanks for your clarifications.

---

### Author Response · Authors · 2024-11-21

**Dear Reviewers,**


Thank you for your insightful comments and feedback, which have been valuable in improving the clarity of our work. In the revised version, we have included additional results on the MNIST dataset for both convex and non-convex settings. Specifically, we have extended our evaluation to include more options of learning rates that are commonly considered for tuning in these setups:


- **Convex setup**: Results are presented for learning rates in $[0.001, 0.002, \ldots, 0.009, 0.01, 0.02, \ldots, 0.09, 0.1, 0.2, \ldots, 0.9]$.
- **Non-convex setup**: Results are presented for learning rates in $[0.01, 0.02, \ldots, 0.09, 0.1, 0.2, \ldots, 0.9]$.


We have also added close-up views for all experiments, including those on CIFAR-10, to illustrate better the process of finding a high-performing learning rate within typical search ranges for each setup. As shown in Figures 1 and 2 in the Experiments section, our approach demonstrates high robustness across a broad range of learning rates. While $\mu^2$-SGD may not always achieve the absolute best result, it consistently achieves high performance with great robustness. This can significantly reduce the need for extensive learning rate tuning—a process that can be computationally expensive.  As a result, the reliability of $\mu^2$-SGD across a wide range of learning rates, including commonly used values, makes it a practical and efficient choice.

---

### Meta-Review · Area_Chair_qUTc · 2024-12-13

**Metareview:**

The paper proposes a new gradient estimator for stochastic convex optimization by combining momentum-based techniques. The reviewer still has some concern on the unusual choices of learning rate. However, all the reviewers recommend the acceptance of the paper. Please incorporate all the discussions and revise the paper accordingly. More discussions about the reviewers' minor comments are needed in the final version.

**Additional Comments On Reviewer Discussion:**

NA

---

### Decision · Program_Chairs · 2025-01-22

Accept (Poster)